# A newly identified gene *Ahed* plays essential roles in murine haematopoiesis

Ritsuko Nakai [1], Takafumi Yokota [1,2] ✉, Masahiro Tokunaga[3,4], Mikiro Takaishi[5], Tomomasa Yokomizo[6], Takao Sudo[1,7], Henyun Shi[1], Yoshiaki Yasumizu [8,9], Daisuke Okuzaki [9,10], Chikara Kokubu[4], Sachiyo Tanaka[4], Katsuyoshi Takaoka[11], Ayako Yamanishi[4], Junko Yoshida [4,12], Hitomi Watanabe[13], Gen Kondoh[13], Kyoji Horie [4,12], Naoki Hosen [1,9,14], Shigetoshi Sano [5] & Junji Takeda [15] ✉

The development of haematopoiesis involves the coordinated action of numerous genes, some of which are implicated in haematological malignancies. However, the biological function of many genes remains elusive and unknown functional genes are likely to remain to be uncovered. Here, we report a previously uncharacterised gene in haematopoiesis, identified by screening mutant embryonic stem cells. The gene, '*attenuated haematopoietic development (Ahed)*', encodes a nuclear protein. Conditional knockout (cKO) of *Ahed* results in anaemia from embryonic day 14.5 onward, leading to prenatal demise. Transplantation experiments demonstrate the incapacity of *Ahed*-deficient haematopoietic cells to reconstitute haematopoiesis in vivo. Employing a tamoxifen-inducible cKO model, we further reveal that *Ahed* deletion impairs the intrinsic capacity of haematopoietic cells in adult mice. *Ahed* deletion affects various pathways, and published databases present cancer patients with somatic mutations in *Ahed*. Collectively, our findings underscore the fundamental roles of *Ahed* in lifelong haematopoiesis, implicating its association with malignancies.

Genes whose functions are essential in early ontogeny often play critical roles in later life. This is particularly true for genes involved in the haematopoietic system in mammals, in which diverse genes play delicate and coordinated roles in a time-dependent manner. Mutations in development-related genes are associated with the occurrence of

haematopoietic neoplasms in adults. For example, a transcription complex encoded by the *core-binding factor-b* (*CBFb*) and *runt-related transcription factor-1* (*Runx1*) genes is imperative for the normal development of bones, neurons, and blood cells[1–3]. Somatic mutations in these genes result in various types of adult haematopoietic

[1]Department of Haematology and Oncology, Graduate School of Medicine, Osaka University, Suita, Osaka 565-0871, Japan. [2]Department of Haematology, Osaka International Cancer Institute, Osaka, Osaka 541-8567, Japan. [3]Department of Haematology, Suita Municipal Hospital, Suita, Osaka 564-0018, Japan. [4]Department of Genome Biology, Graduate School of Medicine, Osaka University, Suita, Osaka 565-0871, Japan. [5]Department of Dermatology, Kochi Medical School, Kochi University, Nankoku, Kochi 783-8505, Japan. [6]Department of Microscopic and Developmental Anatomy, Tokyo Women's Medical University, Shinjuku-ku, Tokyo 162-8666, Japan. [7]Department of Haematology, National Hospital Organisation Osaka National Hospital, Osaka, Osaka 540-0006, Japan. [8]Department of Experimental Immunology, Immunology Frontier Research Centre, Osaka University, Suita, Osaka 565-0871, Japan. [9]Integrated Frontier Research for Medical Science Division, Institute for Open and Transdisciplinary Research Initiatives, Osaka University, Suita, Osaka 565-0871, Japan. [10]Genome Information Research Centre, Research Institute for Microbial Diseases, Osaka University, Suita, Osaka 565-0871, Japan. [11]Developmental Genetics Group, Graduate School of Frontier Biosciences, Osaka University, Suita, Osaka 565-0871, Japan. [12]Department of Physiology II, Nara Medical University, Kashihara, Nara 634-8521, Japan. [13]Laboratory of Animal Experiments for Regeneration, Institute for Frontier Life and Medical Sciences, Kyoto University, Kyoto, Kyoto 606-8507, Japan. [14]Laboratory of Cellular Immunotherapy, World Premier International Immunology Frontier Research Centre, Osaka University, Suita, Osaka 565-0871, Japan. [15]Research Institute for Microbial Diseases, Osaka University, Suita, Osaka 565-0871, Japan. ✉e-mail: yokotat@oici.jp; jjtakeda@biken.osaka-u.ac.jp

malignancies[4–6]. *Gata2* and *Evi1* are also known as pivotal transcription factors for the development and maintenance of haematopoietic stem cells (HSCs), and their mutations strongly induce myelogenous neoplasms[7–11]. Given that the responsible gene mutations are yet to be determined in a subset of patients with blood cancers[12,13], it is important to explore new essential genes in the development of the haematopoietic system. Despite the abundance of enormous amount of available information by the recent advances in DNA sequencing technologies, most genes remain elusive in terms of their biological functions.

We previously constructed a homozygous mutant embryonic stem cell (ESC) bank by introducing gene-trap insertion vectors into properly engineered murine ESCs and then inducing loss of heterozygosity in the cells through the transient suppression of the Bloom syndrome (Blm) gene[14]. In this study, we extend our method to screen for recessive genetic abnormalities involved in the development of haematopoietic cells. Here, we introduce the gene *AU019823*, encoding a previously uncharacterised protein that we reveal to be indispensable for the ontogeny of the haematopoietic system. We name the gene '*attenuated haematopoietic development (Ahed)*', according to its function.

## Results

### Homozygous mutant ESCs screening identified *Ahed*

We previously established a homozygous mutant murine ESC bank[14]. In each ESC clone, a single gene locus underwent bi-allelic disruption by gene-trap cassettes. The number of homozygous mutant clones in the bank reached ~200 (Supplementary Data 1). To discover previously uncharacterised genes in the mutant ESC clones, we focused on 21 genes whose functions remained unknown in the haematopoietic system (Supplementary Table 1). ESC differentiation screening assays were performed using a well-established method of in vitro haematopoietic differentiation, which was supported by co-culture with OP9 stromal cells (Fig. 1a)[15]. Among the 21 homozygous mutant ESC clones tested, we observed 1 mutant clone that showed a severe disruption in haematopoiesis in OP9 stromal cells (Fig. 1b). In this clone, the gene-trap vector was inserted between exons 2 and 3 of the previously uncharacterised gene *AU019823* (NCBI gene: 270156) in a bi-allelic manner (Supplementary Fig. 1a). The gene *AU019823*, located on mouse chromosome 9, was previously annotated by the consensus coding sequence project[16] as a gene encoding an uncharacterised protein of 292 amino acids (a.a.) with no conserved domains. The gene-trap mutant ESC clone differentiated toward foetal liver (FL) kinase-1+ (Flk-1+) early mesodermal cells with an efficiency comparable to that of the parental wild-type (WT) ESCs (Supplementary Fig. 1b). However, these mutant cells showed a dramatic reduction in capacity for haematopoiesis, generating ~70-fold fewer mature haematopoietic colonies than the parental ESCs (Fig. 1c–e). We named this gene *attenuated haematopoietic development*, hereafter referred to as *Ahed*.

Gene-trap mutations in our ESC bank-derived cell lines could be reverted by flippase (Flp) -mediated removal of the gene-trap cassette flanked by a pair of *FRT* sites (Supplementary Fig. 1a). Therefore, we transfected the *Ahed^{m/m}* homozygous gene-trap mutant ESCs with an Flp expression vector to isolate the *Ahed^{m/r}* heterozygous and *Ahed^{r/r}* homozygous revertant ESCs. These were then processed for the OP9 co-culture experiments. Remarkably, the *Ahed^{m/r}* and *Ahed^{r/r}* revertant ESCs showed restoration of the haematopoietic potential at the *Ahed^{wt/wt}* level when induced to undergo myeloid/erythroid development in vitro (Fig. 1c–e). To rule out the possibility that the gene-trap insertion within the *Ahed* locus might affect neighbouring genes, we transfected *Ahed^{m/m}* homozygous mutant ESCs with an *Ahed* cDNA expression vector and processed them in the OP9 co-cultures. Consequently, the exogenous expression of *Ahed* restored blood cell production from *Ahed^{m/m}* mutant ESCs to the WT level (Fig. 1f–h). These results indicate that the *Ahed*

gene plays a non-redundant role in haematopoiesis after the Flk-1+ early mesodermal cell differentiation.

### *Ahed* gene product is a nuclear protein

To understand the biological role of the *Ahed* gene, we examined the subcellular localisation of its gene products. We genetically tagged these products at either the N- or C-terminus with enhanced green fluorescent protein (EGFP) to observe these fusion proteins in *Ahed^{m/m}* ESCs. Confocal fluorescence microscopy showed that EGFP-AHED or AHED-EGFP fusion proteins were localised in the nuclei (Fig. 2a). The expression of *EGFP-Ahed* or *Ahed-EGFP* rescued the deleterious effect of the *Ahed* mutation on haematopoietic cell production as efficiently as that of *Ahed^{wt/wt}*, indicating that these fusion proteins retained biological functions (Supplementary Fig. 2a–b). Moreover, their nuclear localisation remained after ESCs differentiated into PECAM-1+ endothelial cells (Supplementary Fig. 2c). This result was further supported by the immuno-fluorescence analysis of FLAG-tagged Ahed proteins expressed in *Ahed^{m/m}* ESCs (Supplementary Fig. 2d).

Despite the nuclear localisation of the Ahed protein, no typical nuclear localisation signals (NLSs), such as PKKKRRV in SV40 large T antigen and KRX$_{(10)}$KKKK in nucleoplasmin[17,18], were observed in the protein sequence. Instead, a total of eight segments of a loose NLS consensus sequence K(K/R)X(K/R)[17] were present within the domain between a.a. residues 151–260 (Fig. 2b). To determine the functionality of this 110-a.a. domain, we deleted it from the EGFP-*Ahed* construct, which resulted in the diffuse distribution of this fusion protein in both the cytoplasm and nucleus (Fig. 2c). Moreover, a fusion of this 110-a.a. domain with EGFP was sufficient to retain it inside the nucleus (Fig. 2c). The deletion of either the first four or latter four kelvin (K/R)X(K/R) segments from the EGFP-*Ahed* fusion protein affected nuclear localisation (Fig. 2d), suggesting that at least some of these atypical NLSs cooperate with each other.

### *Ahed* deletion causes embryonic demise

According to the BioGPS database (Supplementary Table 2), the *Ahed* gene is ubiquitously expressed whereas its expression levels are generally higher in haematopoietic lineages than in other tissues. We also confirmed the substantial expression of *Ahed* in the haematopoietic stem/progenitor fraction in adult bone marrow (BM) (Supplementary Fig. 3a). Given that the conventional *Ahed* KO mouse died before the onset of haematopoiesis, we generated *Ahed* conditional KO (cKO) mice in which *Ahed* exon 4 was flanked by two *loxP* sequences (Supplementary Fig. 3b–c). Two independent mouse lines, *Ahed^{flox/flox}* (*Ahed^{fl/fl}*) homozygous floxed mice and *Ahed^{fl/+}* heterozygous mice, were viable, fertile, and showed no detectable abnormalities in their gross appearance (Supplementary Fig. 3d) or peripheral blood counts (Supplementary Fig. 3e). As a haematopoietic lineage-specific Cre-deleted line, we used *Vav1-cre* transgenic mice[19,20]. Although the precise expression timing of the *Vav1* promoter is controversial[21], a previous study demonstrated that this promoter becomes activated on $11 \pm 0.5$ embryonic days in emerging HSCs at the aorta-gonad mesonephros region, vitelline and umbilical arteries, and FL[22]. We bred *Ahed^{fl/fl}* floxed mice to *Vav1-cre Ahed^{fl/+}* mice to obtain *Vav1-cre Ahed^{fl/fl}* mice, hereafter referred to as *Ahed* cKO mice. Although the genotype distribution of foetuses was in accordance with the expected Mendelian ratio until E14.5, no live pups were born (Fig. 3a). *Ahed* cKO embryos were normal in gross appearance until E12.5 (Fig. 3b), but became anaemic from E14.5, characterised by FL atrophy (Supplementary Fig. 3f), and finally showed subcutaneous oedema at E16.5 (Fig. 3c), possibly explaining the reason for their perinatal lethality. These results suggest that *Ahed* is indispensable in haematopoietic development to sustain the growth of foetuses.

To determine the significance of *Ahed* in embryonic haematopoiesis, we analysed the E14.5 FL of *Ahed* cKO and their littermate

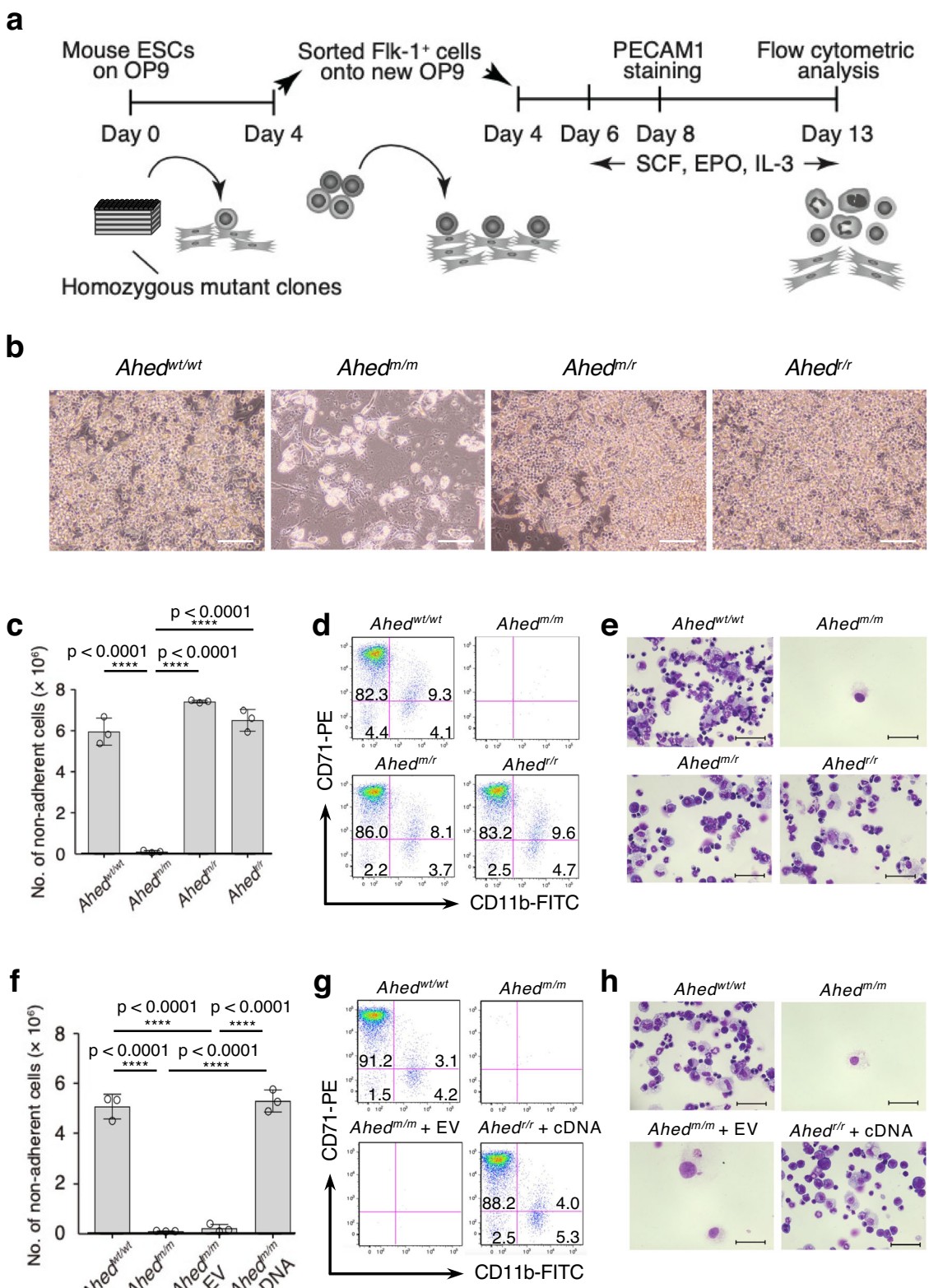

control embryos. Flow cytometric analysis revealed that the relative ratio and the absolute number of TER119⁺-erythroid cells decreased significantly in *Ahed* cKO FLs (Fig. 3d). According to the morphological criteria of erythroid differentiation[23], *Ahed* cKO FL cells showed a decrease in the proportion of acidophilic erythroblasts and denucleated red blood cells (i.e., mature erythroid cells) and an increase in the proportion of pro-erythroblasts and basophilic

erythroblasts (i.e., immature erythroid cells) with no apparent dysplasia (Fig. 3e). This observation was compatible with flow cytometric profiles, showing that the pro-erythroblast (TER119$^{-/Lo}$CD71$^{Hi}$) population was enlarged, whereas that of TER119$^{Hi}$CD71$^{Hi}$ mature erythroid progenitors was reduced in *Ahed* cKO FLs (Fig. 3f). These results suggest that *Ahed* is needed for erythroid maturation at the pro-erythroblast level.

**Fig. 1 | Homozygous mutant ESCs screening identified *Ahed*. a** Procedure of screening homozygous mutant mouse ESC lines by deriving haematopoietic (myeloid and erythroid) cells on OP9 stromal cells. SCF, stem cell factor; EPO, erythropoietin; IL-3, interleukin 3; PECAM-1, platelet/endothelial cell adhesion molecule 1. **b** Phase contrast images of cells on day 10 of ESC differentiation, exhibiting proliferating haematopoietic cells except for in the *Ahed*^m/m culture. Sparkling particles observed in *Ahed*^m/m culture are lipid droplets present in OP9-derived adipocytes (original magnification, × 200). *Ahed*^wt/wt, parental wild-type ESCs; *Ahed*^m/m, homozygous gene-trap mutant ESCs; *Ahed*^m/r, heterozygous revertant ESCs; *Ahed*^r/r, homozygous revertant ESCs. Scale bars, 300 μm. Results shown are representative of three independent experiments. **c** The numbers of non-adherent (blood) cells on day 13 (*n* = 3, biologically independent samples).

**d** Flow cytometric analysis of haematopoietic cells harvested on day 13 of culture. Percentages of cells in each quadrant are shown. Note that the elapsed time in applying each sample was set to be equal. **e** May-Grünwald/Giemsa-stained cyto-centrifuge preparations from each culture on day 13. Scale bars, 50 μm. **f** The numbers of haematopoietic cells on day 13 (*n* = 3, biologically independent samples). EV, empty vector; cDNA, *Ahed* cDNA. **g** Flow cytometric analysis of haematopoietic cells harvested on day 13 of each culture. **h** May-Grünwald/Giemsa-stained cytocentrifuge preparations from each culture on day 13. Scale bars, 50 μm. Data are presented as mean ± standard deviation (s.d.). Statistical significance in **c**, **f** was determined by one-way ANOVA with Brown-Forsythe test. ****p < 0.0001. Source data are provided as a Source Data file.

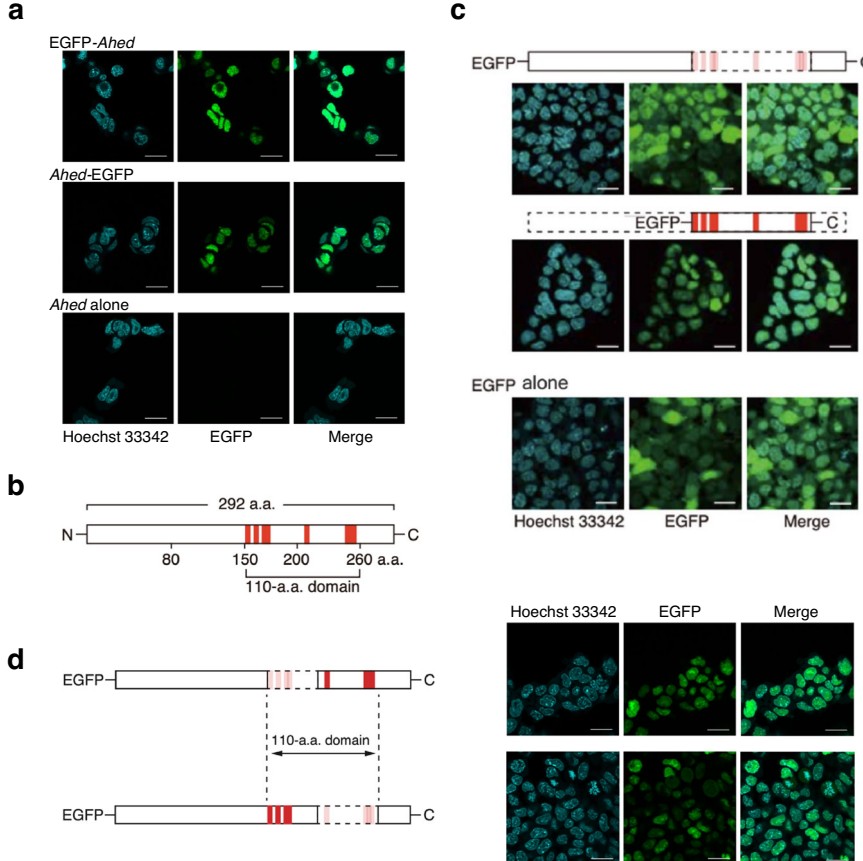

**Fig. 2 | The *Ahed* gene product is a nuclear protein. a** Confocal microscopy images of *Ahed*^m/m ESCs expressing *EGFP*-tagged full length *Ahed* cDNA. *EGFP-Ahed*, *Ahed* cDNA tagged with *EGFP* at the 5′-end; *Ahed-EGFP*, *Ahed* cDNA tagged with *EGFP* at the 3′-end. Nuclei are counterstained with Hoechst 33342. Note that EGFP fluorescence is localised in the nucleus. Scale bars, 20 μm. Results shown are representative of three independent experiments. **b** Schematic of the AHED protein. Red boxes indicate K(K/R)X(K/R) sequences, which are distributed within the domain between a.a. residues 151–260. N, N-terminus; C, C-terminus. **c** Deletion of a.a. 151–260 resulted in EGFP fluorescence in both cytoplasm and nuclei (upper).

Conversely, a.a. residues 151–260 are sufficient for nuclear localisation of the fusion protein (middle). Diffuse distribution of the EGFP protein is shown as a control (lower). Scale bars, 20 μm. **d** Confocal microscopy images of *Ahed*^m/m ESCs expressing *EGFP*-tagged deletion mutants (upper, deletion of a.a. 151–200; lower, deletion of a.a. 201–260). Nuclei are counterstained with Hoechst 33342. Apart from mitotic cells, EGFP fluorescence is localised in the nucleus. Red boxes in the left diagrams indicate K(K/R)X(K/R) sequences. Scale bars, 20 μm. Results shown are representative of three independent experiments.

### *Ahed* is crucial for HSPC differentiation in the foetal liver

Notably, *Ahed* cKO FL cells showed increased numbers and ratios of Lineage⁻Sca-1⁺c-Kit^high (LSK) cells, which represent FL haemato-poietic stem/progenitor cells (HSPCs) (Fig. 4a). Real-time quantitative PCR (RT-qPCR) analysis detected almost no floxed *Ahed* segments in both LSK and TER119⁺ fractions of the *Ahed* cKO FL cells (Supplementary Fig. 4a), suggesting that the increase in LSK cells was not due to a possible bias in Cre-mediated recombination frequency. Due to the discrepancy between the increased number of immature LSK cells and the reduced number of mature blood cells,

flow cytometric analyses were conducted to determine the stage at which the differentiation of HSPCs stagnated. We found that the proportion of HSCs and multipotent progenitors (MPPs), which was phenotypically defined by the expression of cell surface molecules, increased whereas that of lymphoid-primed multipotent progenitors (LMPPs) and common lymphoid progenitors (CLPs) decreased in *Ahed* cKO FLs. The calculated cell counts showed the remarkable gap between MPPs and LMPPs (Fig. 4b), indicating that the early stage of haematopoietic differentiation toward the lymphoid lineage was detrimentally affected by the loss of *Ahed*. On

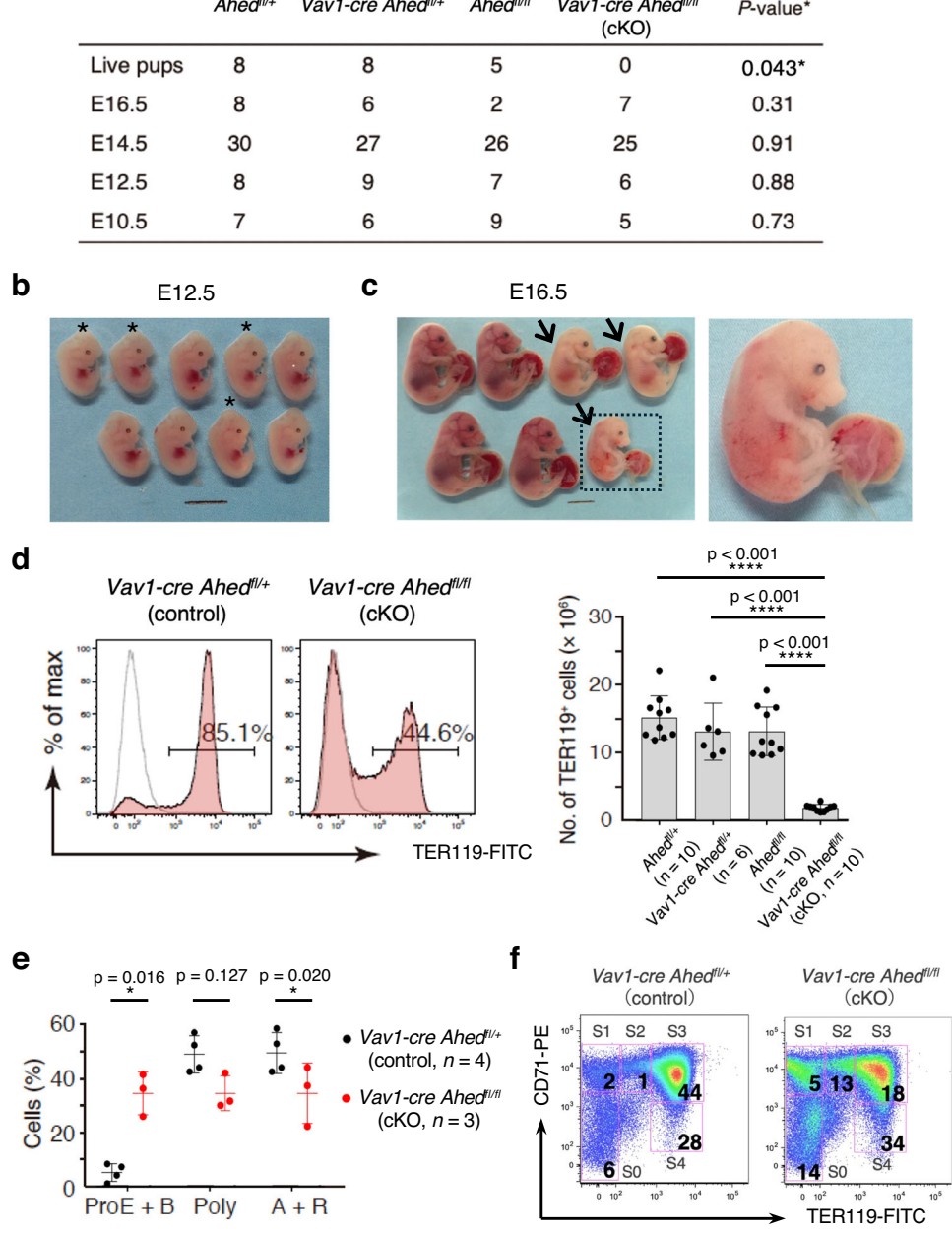

**Fig. 3 | *Ahed* deletion causes severely anaemic and embryonic demise.**
**a** Genotyping results of the pups and embryos obtained from *Ahed*[+/−]; *Vav1-cre* and *Ahed*[fl/fl] pairs at the indicated points. **b** Embryos recovered at E12.5. Asterisks indicate the *Ahed* cKO (*Vav1-cre Ahed*[fl/fl]) embryos. Scale bars, 5 mm. **c** Embryos recovered at E16.5 (left). Arrows indicate *Ahed* cKO embryos. Scale bar, 5 mm. The embryo in the dotted square exhibits subcutaneous oedema (arrowhead, right). **d** E14.5 foetal liver cells were stained with an isotype control IgG (dashed line) or an anti-TER119 Ab (solid line) and analysed by flow cytometry. Representative data for the indicated genotypes are shown. Absolute numbers of TER119[+] erythroid cells in E14.5 foetal liver were calculated from the flow cytometric data. Horizontal lines indicate median values. **e** Graph showing the proportion of E14.5 foetal liver erythroid cells in each developmental stage. ProE, proerythroblasts; B, basophilic cells;

Poly, polychromatic cells; A, acidophilic cells; R, denucleated red blood cells. **f** Flow cytometry plots of E14.5 foetal liver erythrocytes from *Vav1-cre Ahed*[fl/+] (control) and *Vav1-cre Ahed*[fl/fl] (cKO) mice to evaluate erythroid differentiation using Ter119 and CD71. Each subset group was defined as follows; S0, TER119[−]CD71[−]; S1, TER119[−]CD71[+]; S2, TER119[Lo]CD71[+]; S3, TER119[+]CD71[+]; S4, TER119[−]CD71[+]. Numbers indicate the percentage of each fraction. Results shown are representative of three independent experiments. Data are presented as mean ± s.d. Statistical significance in **a** was determined by one-sided Chi-square test, *against expectations of Mendelian segregation, in **d**, **f** by one-way ANOVA with Brown-Forsythe test, and in **e** by two-sided unpaired Student's *t*-test. *p < 0.05; ****p < 0.0001; NS, not significant. Source data are provided as a Source Data file.

the contrary, the proportion and number of common myeloid progenitors (CMPs), granulocyte-monocyte progenitors (GMPs), and megakaryocyte-erythroid progenitors (MEPs) was sustained (Fig. 4c), but more mature precursors bearing linage-related markers such as CD11b or TER119 significantly decreased (Fig. 3d, f). Thus, it was evident that the differentiation potential of HSPCs

deteriorated without *Ahed* at early stages in all lineage directions (Fig. 4d).

Next, we assessed the growth and differentiation potential of HSPCs derived from E14.5 *Ahed* cKO FLs. The methylcellulose assay revealed that E14.5 *Ahed* cKO FL LSK cells produced fewer and smaller (< 100 cells) colonies than the control cells, indicating their impaired

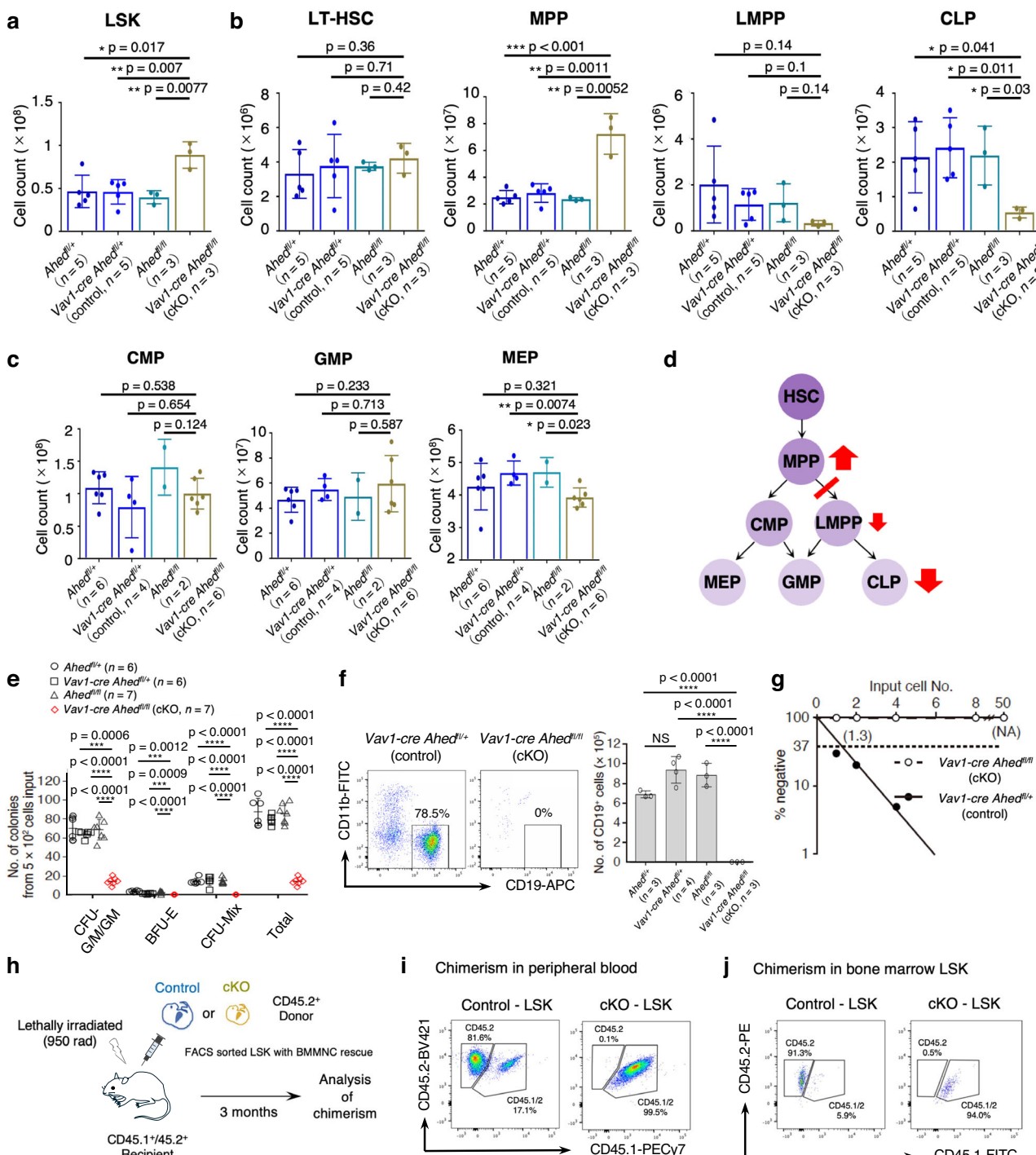

**Fig. 4 | *Ahed* is indispensable for HSPC differentiation. a** Absolute numbers of LSK cells in E14.5 foetal liver (FL). **b** Absolute numbers of LT-HSCs, MPPs, LMPPs, and CLPs ($n = 3$–5), pooled from two independent experiments. In bar charts, results are shown as mean ± s.d. and each dot represents an individual FL result. **c** Absolute numbers of CMPs, GMPs, MEPs ($n = 2$–6). **d** A schematic representation summarizing (**a**–**c**). **e** LSK cells from E14.5 FL were subjected to methylcellulose culture. Generated colonies were classified and counted. BFU-E, burst-forming units-erythroid; CFU-G/M/GM, colony-forming unit-granulocyte/macrophage/granulocyte-macrophage. **f** E14.5 FL LSK cells were co-cultured with MS-5 in the presence of SCF, Flt3-ligand and IL-7. Flow cytometry profiles of recovered CD45$^+$ cells (left). Absolute numbers of CD19$^+$ B cells differentiated from $2.5 \times 10^2$ LSK cells (right). **g** Limiting dilution analysis of E14.5 FL LSK cells using MS-5 co-culture. Frequencies of haematopoietic progenitor cells were determined and shown in

parenthesis. NA, not analysed. **h** Transplantation strategy. Donor *Vav1-cre Ahed*$^{fl/+}$ (control) or *Vav1-cre Ahed*$^{fl/fl}$ (cKO) E14.5 LSK cells sorted from CD45.1$^-$CD45.2$^+$ and transplanted into lethally irradiated CD45.1$^+$CD45.2$^+$ WT mice ($n = 7$–8) with $5 \times 10^5$ CD45.1$^+$CD45.2$^+$ WT BM cells as a rescue. **i** Chimerism of peripheral blood in recipients. Numbers in each figure indicate percentages of each fraction. Results shown are representative of two independent experiments. **j** Chimerism of LSK in recipients' bone marrow. Numbers in each figure indicate percentages of each fraction. Results shown are representative of two independent experiments. Data are presented as mean ± s.d. Statistical significance in **a**-**c**, **i**, **j** was determined by one-way ANOVA with the Tukey–Kramer post-hoc test, and in **e**, **f** by one-way ANOVA with Brown-Forsythe test. *$p < 0.05$; **$p < 0.01$; **$p < 0.001$; ****$p < 0.0001$. NS, not significant. Source data are provided as a Source Data file.

capacity for myeloid and erythroid development (Fig. 4e). We also co-cultured LSK cells with the murine stromal cell line MS-5[24] to assess the growth of B-lineage cells[25]. *Ahed* cKO FL LSK cells were incapable of generating CD19[+] B cells (Fig. 4f). Furthermore, we performed a limiting dilution analysis of E14.5 FL LSK cells in co-culture experiments to quantify the frequencies of haematopoietic progenitors[25]. While 77% of control LSK cells had haematopoietic growth potential, none of the *Ahed* cKO LSK cells showed progenitor capacity (Fig. 4g). Taken together, *Ahed* cKO-derived LSK cells lack the ability to produce mature haematopoietic colonies in vitro.

Then, we conducted transplantation experiments to evaluate the long-term haematopoietic capability of *Ahed* cKO LSK cells; 1000 FL control or *Ahed* cKO LSK cells of the CD45.1[−]CD45.2[+] phenotype were transplanted along with $5 \times 10^5$ CD45.1[+]CD45.2[+] WT BM cells as a rescue to lethally irradiated CD45.1[+]CD45.2[+] WT mouse (Fig. 4h). Eight weeks after transplantation, we detected approximately 80% of chimerism of CD45.1[−]CD45.2[+] WT LSK cells in the recipient peripheral blood, whereas we observed essentially no contribution of *Ahed* cKO cells to the recovered leucocytes (Fig. 4i). Furthermore, BM examinations 12 weeks after the transplantation determined no contribution of *Ahed* cKO LSK-derived haematopoietic cells in any lineages, including the most primitive HSC fraction (Fig. 4j). These results indicate that *Ahed* is indispensable to maintain the integrity of HSPCs in the FL by sustaining the engraftment and haematopoietic-reconstituting potential in vivo.

### *Ahed* is unessential for endothelial-to-haematopoietic transition
Although *Ahed* was dispensable for ESCs to produce Flk-1-positive mesodermal cells, it was unclear whether its deletion affects the formation of haematopoietic endothelium or endothelium to haematopoietic transition. Thus, we generated cKO mice by crossing *Ahed*-floxed mice with *Tie2-cre* transgenic strain, which induce *Ahed*-deletion in both endothelial and haematopoietic lineages[26]. While the developmental defect of these cKO mice became fatal around E14.5 (Supplementary Fig. 4b), their size and appearance were normal at E10.5 (Supplementary Fig. 4c). Fluorescence-activated cell sorter (FACS) analysis of the caudal half region containing vitelline artery (VA) and umbilical artery (UA) (Supplementary Fig. 4d, f) or yolk sac (Supplementary Fig. 4e) of E10.5 embryos revealed that the populations of both c-Kit[+]CD31[+] emerging HSCs and c-Kit[+]CD31[+]CD45[−] haemogenic endothelial cells were intact in *Ahed* cKO embryos (Supplementary Fig. 4d–f). Three-dimensional analysis of aortic haemopoietic clusters[11] was also performed to examine the transition from endothelium to haematopoietic cells in detail. Similar to the FACS data, the number and formation of haemogenic clusters in the *Ahed* cKO dorsal aorta was not inferior to those in controls (Supplementary Fig. 4g, h). Collectively, we conclude that *Ahed* is unessential for haematopoietic endothelial formation and endothelial-to-haematopoietic transition.

### *Ahed* plays an indispensable role in adult haematopoiesis
We wondered whether *Ahed* also contributes to homoeostatic haematopoiesis in the BM after birth. To investigate *Ahed*'s role post-natally in vivo, we generated a tamoxifen-inducible *Ahed* cKO model by crossing *Ahed^{fl/fl}* mice with Cre recombinase-oestrogen receptor T2 (Cre-ER[T2]) expressing mice under the control of the *Rosa26* gene promoter. Tamoxifen injection for 5 consecutive days efficiently deleted the *Ahed* gene from BM cells of *Rosa26-CreER^{T2}* (*R26CE*) *Ahed^{fl/fl}* mice (Fig. 5a, Supplementary Fig. 5a). Tamoxifen-treated *R26CE Ahed^{fl/fl}* mice lost weight precipitously after day 10 and began to die at approximately day 14. Thus, analyses for haematopoiesis were performed on day 10 (Fig. 5a). We observed that the cellularity of BM was severely reduced in tamoxifen-treated *R26CE Ahed^{fl/fl}* mice compared with that in tamoxifen-treated *R26CE Ahed^{fl/+}* control mice (Fig. 5b). Accordingly, the tamoxifen-treated *R26CE Ahed^{fl/fl}* mice developed cytopenia, particularly lymphocytopenia, in peripheral blood (Fig. 5c). The number of LSK cells were sustained in the BM of tamoxifen-treated *R26CE Ahed^{fl/fl}*

mice, similar to *Ahed* cKO FLs (Supplementary Fig. 4b). However, the myeloid-erythroid colony forming capability was significantly impaired by the deletion of the *Ahed* gene (Fig. 5d). Flow cytometric analysis showed that the stagnation of erythroid differentiation at the pro-erythroblast level occurred in tamoxifen-treated *R26CE Ahed^{fl/fl}* mice (Fig. 4e). We also observed an increased number of phenotypical MEPs, whereas the number of CMPs and GMPs was sustained in the cKO BM (Supplementary Fig. 5c).

In addition to severe hypoplasia of the BM, we detected various abnormalities in non-haematopoietic organs of tamoxifen-treated *R26CE Ahed^{fl/fl}* mice, compared with those of tamoxifen-treated *R26CE Ahed^{fl/+}* control mice. Epidermal atrophy was observed in the skin, and mucosal epithelium of the jejunum was degenerated and infiltrated with inflammatory cells (Supplementary Fig. 5d). Besides, the epithelium of the tongue and oesophagus showed loss of the granular layer, and the pancreas showed a clear decrease in secretory granules in the epithelium of the adenohypophysis (data not shown). These observations suggest that while *Ahed* plays essential roles in lympho-haematopoiesis, it is also involved in the homoeostasis of diverse organs.

The induced *Ahed* knockout after birth causes abnormalities in multiple organs and cell lineages. To clarify whether the role of *Ahed* in postnatal homoeostatic haematopoiesis is self-intrinsic or mediated by non-haematopoietic cells, LSK transplantation studies were conducted using postnatal bone marrow cells from control (tamoxifen-treated *Ahed^{fl/fl}*) or *Ahed* cKO (tamoxifen-treated *R26CE Ahed^{fl/fl}*) mice. $2 \times 10^3$ control or *Ahed* cKO LSK cells of the CD45.2[+] phenotype were transplanted along with $2 \times 10^5$ CD45.1[+] WT BM cells as a rescue to the lethally irradiated CD45.1[+] WT mice (Fig. 5f). Three months after the transplantation, we found that contribution of *Ahed* cKO LSK cells was meager in the peripheral blood (Fig. 5g) and BM (Supplementary Fig. 5e).

Next, to exclude the influence of *Ahed* deletion on the haematopoietic environment of BM, generally called 'HSC miche', an additional experiment was performed; here, LSK cells were sorted from BM mononuclear cells treated with 4-hydroxytamoxifen (4-OHT) for 54 hrs and were transplanted to recipients (Fig. 5h). In this experimental setting, we determined that the *Ahed*-deleted LSK cells essentially lost their haematopoietic capacity. Three months after the transplantation, the donor cell chimerism transplanted with 4-OHT-treated *R26CE Ahed^{fl/fl}* LSK cells was significantly lower in peripheral blood (Fig. 5i) than control group. Additionally, PCR examinations revealed that an incompletely deleted *Ahed^{fl/fl}* band (247 bp) was detected in the small CD45.2[+] population of peripheral blood reconstituted from the 4-OHT-treated *R26CE Ahed^{fl/fl}* LSK cells whereas a KO band at 183 bp was invisible (Fig. 5i and Supplementary Fig. 5f). Paradoxically, this observation suggested that LSK cells with intact *Ahed* were competent to give rise to haematopoietic cells. Collectively, *Ahed* plays a critical role in the haematopoietic system throughout the life in mice. It is indispensable for the differentiation of HSPCs, and its loss of function is disruptive to normal haematopoiesis.

### *Ahed* deficiency induces cellular apoptosis
According to the observations showing that Ahed-deleted LSK cells were hampered in the growth and differentiation, we first examined the expression levels of cell-cycle regulating gens by RT-PCR. Consequently, we detected upregulation of p21 expression, one of the major targets of p53, by Ahed deletion, which increased 1.22-fold in FL and 2.83-fold in adult BM HSCs (Supplementary Table 4). Other target genes of p53, such as *Prep*, were also upregulated, suggesting that cell cycle arrest and cellular apoptosis might be involved in the differentiation block of *Ahed*-deficient HSPCs. To test this possibility, we cultured *Ahed* cKO FL- and control FL-derived LSK cells in a differentiation medium. After 3 and 5 days culture of the FL-derived LSK cells, *Ahed* cKO-derived cells showed a significant decrease in cell number (Fig. 6a and Supplementary Fig. 6a). The percentage of

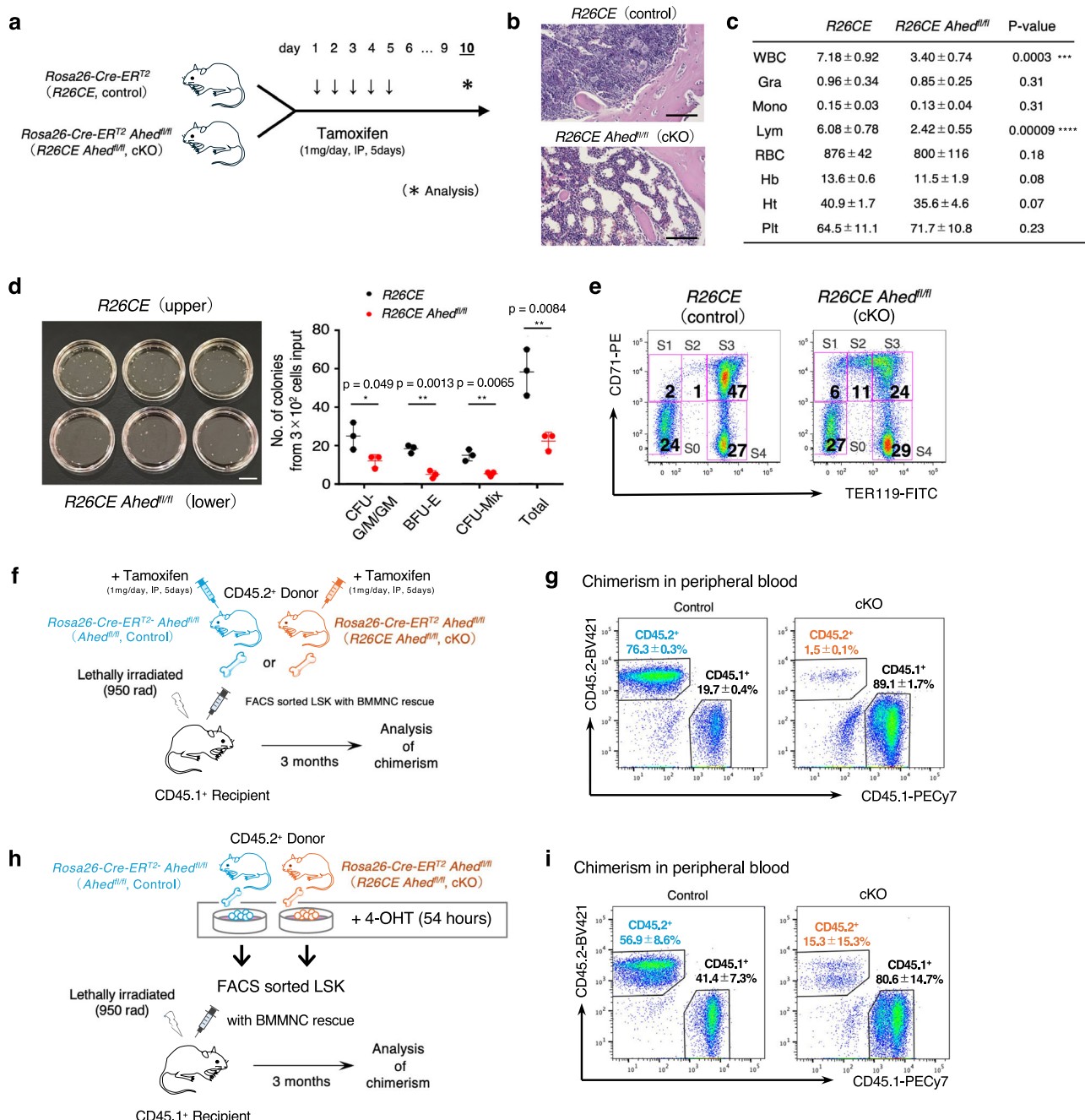

**Fig. 5 | *Ahed* plays essential roles in adult haematopoiesis. a** Schematic of the procedure. Tamoxifen injection to *Rosa26-CreERT2 Ahed*[fl/+] (*R26CE*; control) and *Rosa26-CreERT2 Ahed*[fl/fl] (*R26CE Ahed*[fl/fl]; cKO) mice for 5 consecutive days and analysis of BM on day 10. **b** Hematoxylin and eosin (H&E) staining of bone marrow tissues from *R26CE* and *R26CE Ahed*[fl/fl] mice. Scale bars, 100 μm. Results shown are representative of two independent experiments. **c** Peripheral blood counts; *n* = 3, in *R26CE* (control) and *n* = 6, in *R26CE Ahed*[fl/fl] (cKO). **d** A total of 3.0 × 10² LSK cells from BM were subjected to methylcellulose culture for 9 days. The numbers of colonies are shown in a right graph. Scale bars, 1 cm. **e** Flow cytometry plots of E14.5 foetal liver erythrocytes from *R26CE* and *R26CE Ahed*[fl/fl] mice to evaluate erythroid differentiation using TER119 and CD71. Subset groups were defined as follows; S0, TER119⁻CD71⁻; S1, TER119⁻CD71⁺; S2, TER119[Lo]CD71⁺; S3, TER119⁺CD71⁺; S4, TER119⁻CD71⁺. Numbers indicate the percentage of each fraction. **f** Transplantation strategy. Donor *Ahed*[fl/fl] (control) or *R26CE Ahed*[fl/fl]

(cKO) BM LSK cells sorted from CD45.1⁻CD45.2⁺ and transplanted into lethally irradiated CD45.1⁺CD45.2⁻ WT mice (*n* = 4 for both group) with CD45.1⁺CD45.2⁻ WT BM cells as a rescue. **g** Chimerism of peripheral blood in recipients. Numbers in each figure indicate percentages of each fraction. **h** Transplantation strategy. Donor *Ahed*[fl/fl] (control) or *R26CE Ahed*[fl/fl] (cKO) BM LSK cells were sorted by FACS from CD45.1⁻CD45.2⁺ BM mononuclear cells treated with 5 μM 4-OHT for 54 hours and transplanted into lethally irradiated CD45.1⁺CD45.2⁻ WT mice (*n* = 4 for control and *n* = 8 for cKO group). **i** Chimerism of peripheral blood in recipients. Numbers in each figure indicate percentages of each fraction. Data are presented as mean ± s.d. Statistical significance in **c** was determined by two-sided unpaired Student's *t*-test, in **d** by one-way ANOVA with the Tukey–Kramer post-hoc test, and in **e, g, i** by one-way ANOVA with Brown-Forsythe test. *p < 0.05; **p < 0.01, ***p < 0.001, ****p < 0.0001. Source data are provided as a Source Data file.

proliferating cells also decreased (Supplementary Fig. 6b). Annexin V staining revealed that the number of apoptotic cells increased in both the early and late phases (Fig. 6b). Furthermore, *Ahed* cKO-derived cells included a higher subdiploid proportion and a lower proportion

of the S-G2/M fraction compared with control-derived cells (Fig. 6c). Taken together, *Ahed* deletion affects the p53 pathway in early haematopoiesis, resulting in the differentiation block, at least in part, via inducing cell cycle arrest and apoptosis.

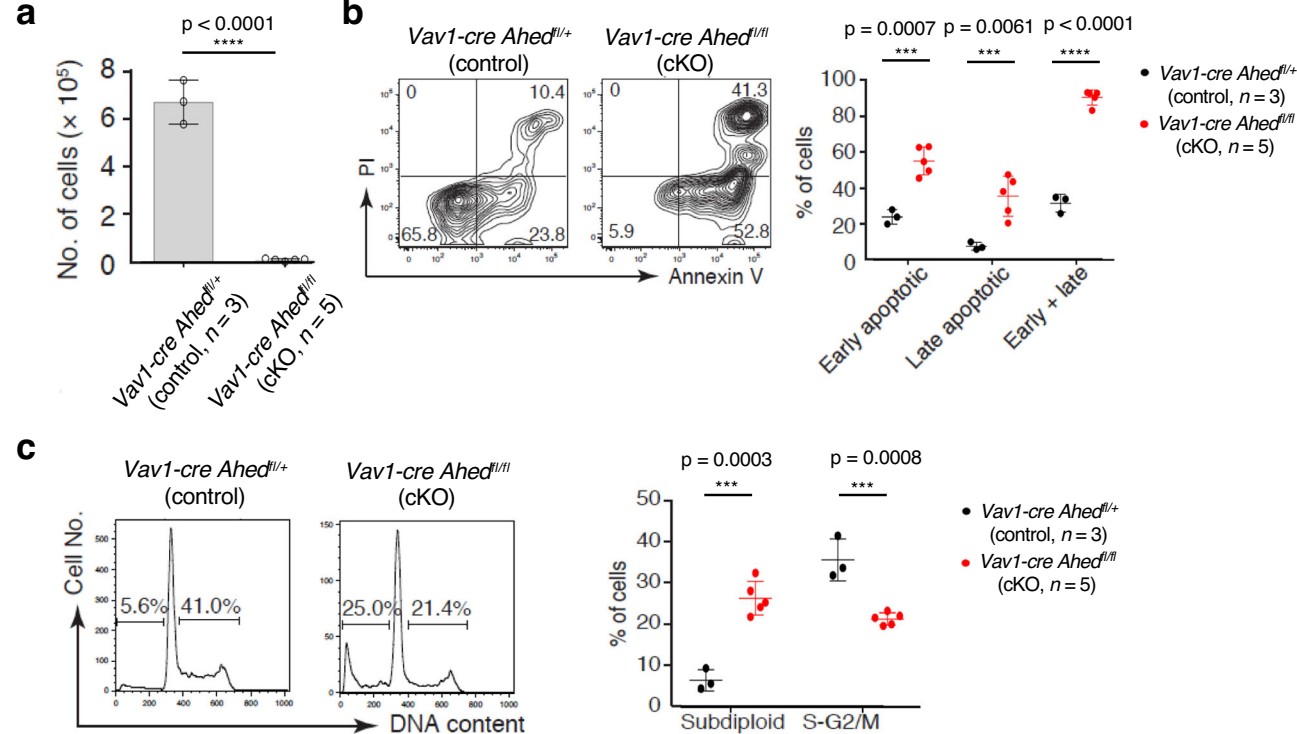

**Fig. 6 | Ahed deficiency induces cellular apoptosis. a** Number of cells yielded on day 5 culture of the FL-derived LSK cells. **b** Flow cytometry contour plots showing live (PI⁻, Annexin V⁻), early apoptotic (PI⁻, Annexin V⁺), late apoptotic (PI⁺, Annexin V⁺), and necrotic (PI⁺, Annexin V⁻) cell fractions on day 5 culture of the FL-derived LSK cells (left). The CD11b⁺ fraction was analysed. Percentages of cells in each quadrant are shown (right). **c** DNA content of cultured cells on day 5 culture of the

FL-derived LSK cells (left). The proportions of apoptotic cells (DNA content < 2n) and cells in S-G2/M phase are shown (right). Data are presented as mean ± s.d. Statistical significance in **a**, **c** was determined by two-sided unpaired Student's t-test, and in **b** by one-way ANOVA with Brown-Forsythe test. ***p < 0.01, ****p < 0.0001. Source data are provided as a Source Data file.

## Pan-haematopoietic *Ahed* deletion disrupts critical gene expression

Next, we conducted RNA-sequencing (RNA-seq) analyses to compare the gene expression features of haematopoietic cells derived from FLs and adult BM of control or *Ahed*-deficient mice. In these experiments, the LSK CD48⁻ HSC-enriched fraction was sorted to determine early changes in haematopoiesis. We postulated that the genes whose expression patterns were altered in both foetal and adult cells in the same manner would be important because such genes were possibly critical factors for haematopoiesis in both ontogeny and adulthood.

In terms of the *Ahed*-deficient HSC-enriched fraction, we observed that 100 genes were commonly upregulated in both foetal and adult samples, whereas 259 genes were downregulated (Fig. 7a). Principal component analysis (PCA) of FL results revealed that the control and *Ahed* cKO samples were well-separated in PC1 (Supplementary Fig. 7a). Comparative transcriptome analysis was performed to extract the differentially expressed genes (DEGs) (Supplementary Fig. 7b). The study findings revealed that genes involved in the immune response pathway were enriched and downregulated in the *Ahed* cKO FL LSK cells compared with those in controls (Supplementary Fig. 7c). Ingenuity pathway analysis showed that *Ahed* deletion markedly downregulated various signalling pathways such as 'haematological system development' and 'cellular movement' in the HSC-enriched fractions of both FL and adult BM (Fig. 7b and Supplementary Fig. 7d). Upstream regulator analyses revealed that the *Gata2* pathway was most significantly hampered in both FL and adult BM (Fig. 7c). Additionally, we confirmed the downregulation of expression levels of HSC-related genes, such as *Gata2*, *Lmo2*, and *Runx1* in E14.5 FL LSK cells in *Ahed* cKO mice (Fig. 7d and Supplementary Table 3). This suggests that although the number of cells in the HSC-enriched fraction was sustained in *Ahed* cKO-derived FL and adult BM, the biological integrity of these cells was

substantially damaged. Although the underlying molecular mechanisms remain to be determined, it should be noted that somatic mutations of *Ahed* gene have been repeatedly detected in patients with haematopoietic neoplasms such as acute myeloid leukaemia, acute lymphoblastic leukaemia and various types of malignant lymphoma, as well as in many solid tumour cases with hotspots at residues 1ˢᵗ, 6ᵗʰ, and 291ˢᵗ glutamine (Fig. 7e–f).

## Discussion

There have been no comprehensive reports on the analysis of the function and molecular mechanism of *Ahed*, including its human orthologue, *C11orf57*. The reason why an essential factor for haematopoiesis such as *Ahed* has not been discovered is largely due to the fact that all homozygous mutants die *in utero*, whereas heterozygous deficient individuals stay healthy. Therefore, our homozygous mutant ESCs screening is an effective method to identify novel molecules involved in such recessive genetic abnormalities.

It is noteworthy that the number of LSK cells increased in *Ahed* cKO FLs despite the efficient *Vav1-cre*-mediated cleavage of this gene locus after E11 ± 0.5. This observation suggested that HSCs could develop and proliferate in the FL without *Ahed*, although their differentiation potential toward functional blood cells was completely disrupted. Moreover, the LSK fraction was maintained in the *Ahed* cKO FL, but more differentiated progenitor cells such as LMPPs and CLPs were reduced (Fig. 4a–b). We conclude from this result that *Ahed* is mainly involved in the differentiation potential of HSCs, but not in self-renewing proliferation. A recent study reported that erythroid and myeloid progenitors (EMPs) with adult haemoglobin do not originate in authentic HSCs[27] and that authentic HSC-derived CMPs, MEPs, and GMPs account for less than 20% of the total, even at E16.5[28]. Therefore, we infer that most of the CMPs, MEPs, and GMPs in *Ahed* cKO FL at

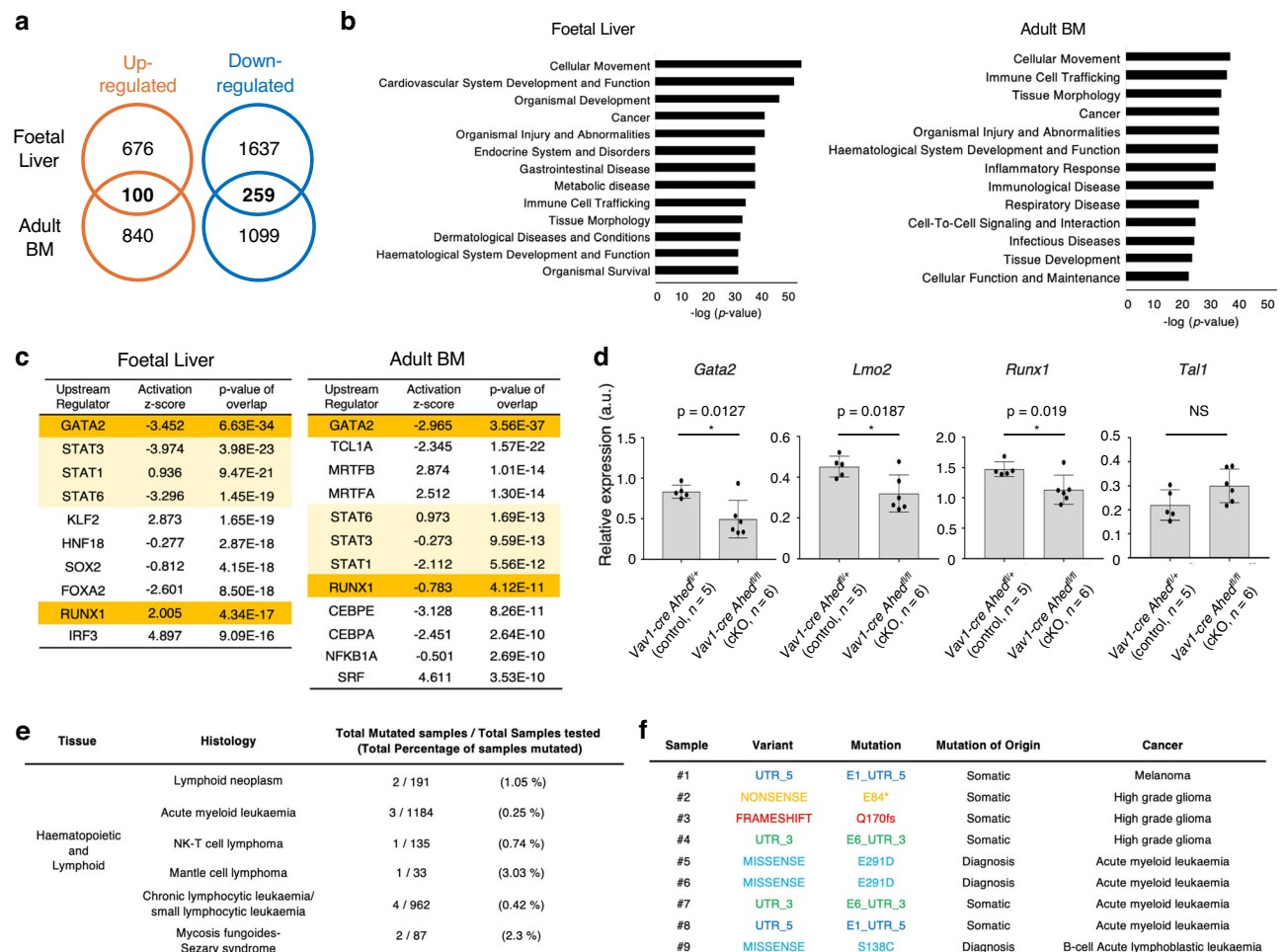

**Fig. 7 | Pan-haematopoietic *Ahed* deletion disrupts critical gene expression.**
**a** Differentially expressed genes from RNA-seq in foetal liver (FL) were compared with those in adult bone marrow (BM), in which gene expressions of *Vav1-cre Ahed^fl/fl* (cKO) versus *Vav1-cre Ahed^fl/+* (control) in E14.5 FL, and *R26CE Ahed^fl/fl* (cKO) versus *R26CE Ahed^fl/+* (control) mice in BM were analysed. A strong positive correlation was observed between the two groups. The Venn diagram shows the number of differentially expressed genes that overlapped between the two groups. **b** The top 10 enriched biological functional categories in which gene expressions of cKO versus control conducted in E14.5 FL (left) and adult BM (right), according to the Ingenuity Knowledge Base. The x-axis shows the significance, which is the value of −log (p value). **c** Upstream transcriptional regulators in *Ahed* cKO FLs compared with control FL (left), and in *R26CE Ahed^fl/fl* (cKO) BM compared with control BM (right). **d** Expression levels of representative haematopoietic stem cell-related genes, *Gata2,*

*Lmo2, Runx1* and *Tal1* in E14.5 foetal liver LSK cells. Data are shown as mean expression levels ± s.d. a.u., arbitrary unit. *Vav1-cre Ahed^fl/+* (n = 5); *Vav1-cre Ahed^fl/fl* (n = 6). **e** List of haematopoietic neoplasm patients with *C11orf57* mutations from the database (COSMIC; https://cancer.sanger.ac.uk/cosmic/gene/analysis); somatic mutations of *Ahed* were found in some patients with haematopoietic neoplasm such as acute myeloid leukaemia, myelodysplastic syndrome, and chronic lymphocytic leukemia. **f** List of patients with *C11orf57* mutations from the database, including splicing variant patterns (St. Jude Cloud PeCan; https://pecan.stjude.cloud/proteinpaint/). Statistical significance in **b**, **c** was determined by the Fisher's Exact Test (conducted with Ingenuity Pathway Analysis software (Ingenuity Systems; Qiagen)), and in **d** by two-sided unpaired Student's *t*-test. *p < 0.05. NS, not significant. Source data are provided as a Source Data file.

E14.5 were supplied by yolk-sac-derived EMPs, resulting in no obvious difference between the control and *Ahed* cKO groups in our analyses (Fig. 4c).

Upstream regulator analyses of both FL and adult BMs identified the *Gata2* and *Runx1* signalling pathway as an important upstream regulator in the transcriptome of *Ahed*-deficient HSPCs (Fig. 7c). RT-PCR tests with specific primers determined that the expressions of several critical genes for HSC function, such as *Gata2*, *Runx1*, and *Lmo2* were concomitantly down-regulated in *Ahed* cKO HSPCs (Fig. 7d). As *Gata2* plays a crucial role in maintaining the integrity of HSCs[29], inhibition of the *Gata2* signalling pathway has a negative impact on haematopoiesis. In the absence of *Ahed*, lymphopoiesis were markedly hampered in both FL and adult BMs (Figs. 4b and 5c). *Runx1*, one of the master genes in haematopoiesis, plays essential roles to sustain the lymphoid lineage via regulating *PU.1*, its main target gene[30,31]. Thus, we infer that the compromised lymphopoiesis in *Ahed*-deficient mice is at

least partly due to the aberrant expression levels of those critical genes.

Based on its nuclear localisation (Fig. 2a–d), we believed that the Ahed protein is involved in intranuclear events. Given that the gene expression analyses above determined the involvement of *Ahed* in critical pathways encompassing various biological processes, we sought the underlying mechanisms of how this protein executes its biological functions. A search through the biophysical interactions in the ORFeome-based complexes 3.0 (BioPlex 3.0) network database indicated a close correlation of Ahed protein in the RNA splicing process (Supplementary Fig. 8a–c)[32–34]. To examine the possible involvement of Ahed protein in alternative splicing in HSPCs, we performed high-throughput RNA-Seq in triplicate comparing control and *Ahed* cKO FL LSK cells. Then, we checked this read depth by making saturation plots for the number of splicing junctions detected in each sample (Supplementary Fig. 8d). Subsequently, we exploited a recently

developed method 'Leafcutter' which permitted us to identify novel as well as known alternative splicing events and to quantify differential intron usage across samples using short-read RNA-Seq data without relying on predefined transcript annotation[35]. Skewed splicing patterns were observed in diverse transcripts for as many as 1369 genes (Supplementary Fig. 8e and Supplementary Table 5), including *PU.1*, *Ythdf2*, *Sel1l*, and *Zfp60* (Supplementary Fig. 8f), which are known to play important roles in haematopoiesis[36–39]. As the PC analysis results showed, the control and *Ahed* cKO samples were well-separated in PC1 (Supplementary Fig. 8g). Then, we synthesised specific primers for each gene and performed conventional reverse transcription PCR (RT-PCR) on control and *Ahed* cKO LSK cells to examine the lengths of the transcripts (Supplementary Fig. 8h and Supplementary Table 6). Consequently, this approach revealed the presence of aberrant transcripts with larger sizes in *Ahed* cKO LSK (Supplementary Fig. 8i). Finally, we determined that intron sequences were retained in transcripts of *Ahed* cKO HSPCs by sequencing those RT-PCR products (Supplementary Fig. 8j). Further enrichment analysis suggested that this abnormality would dysregulate intracellular signal transduction of those cells tremendously, which might underlie the critical role of *Ahed* in sustaining the integrity of HSPCs (Supplementary Fig. 8k). Although direct evidence is lacking at this stage, it can be deduced that the Ahed protein plays a pivotal role in the development of haematopoiesis possibly by coordinating post-transcriptional RNA regulation. The malfunction of this basic and vital process might have damaged the critical pathways including *Gata2* and *Runx1-PU.1* signalling, and finally activated the p21-p53 pathway in HSPCs, resulting in fatally impaired haematopoiesis of *Ahed*-deficient mice.

The mouse *Ahed* gene has orthologues in diverse vertebrate species, including humans. The human ortholog, *C11orf57*, is located on chromosome 11 and encodes a 293-a.a. protein with 83% identity to the mouse protein. Recently, a CRISPR/Cas9-based gene KO screening using five different human cell lines identified genes whose perturbation decreased cell growth and proliferation[40]. The authors referred to these genes as 'fitness' genes; according to the definition, 'essential' genes consist of a subset of fitness genes. Among the genes, 'core fitness' genes that were shared by at least three different cell lines were also defined. Notably, *C11orf57* was included in the list of core fitness genes (The Toronto KnockOut Library, http://tko.ccbr.utoronto.ca/), although this gene was not specifically mentioned in the paper[40]. Given that the five cell lines used were all human cancer or immortalised somatic cell lines, the protein encoded by this gene was thought to serve as a biologically essential element in malignant cells.

It would be challenging to determine the direct association of Ahed protein in the components of the spliceosomes in an extremely small population such as HSCs whereas pre-publication data show that Ahed is likely to associate with PRPF8, SF3B1, and RBM39 in Hela cells (https://assets.researchsquare.com/files/rs-3234334/v1/c544e979-dfba-4119-b164-61a01f4384f3.pdf?c=1699278362). Proper recognition of intron sequences by the spliceosome is essential for intron clearance. Mutations affecting the spliceosome or splicing factors can result in perturbations in downstream splicing targets, many of which are in signalling pathways involved in cancer. Alternatively, copy number or expression levels of splicing factors can be altered in tumours without a mutational antecedent[41]. In recent years, rapid and high-throughput DNA sequencing has revealed genetic mutations of RNA splicing factors in various neoplastic diseases. Regarding haematopoietic malignancies, mutations of *SF3B1* and *SRSF2* are frequently observed in myeloid neoplasms such as myelodysplastic syndrome[42,43]. *SF3B1* and *SRSF2* mutations occur at specific amino acid residues, so-called hotspots, which affect their protein conformation and result in gain or change-of-function for RNA splicing[44]. Consequently, aberrant and potentially oncogenic products are educed to arise due to mis-splicing. The BioPlex network database shows the association of Ahed protein with various splicing factors including *SF3A* and *SRSF* proteins (Supplementary Fig. 8a), implying that *Ahed* mutations would hamper the proper maturation of mRNAs, which is also likely to underlie the development of haematopoietic malignancies. Further studies are needed to determine the pathological significances of missense or UTR mutations of *Ahed* gene, which have been repeatedly observed in cancer patients (Fig. 7f).

In conclusion, we report the first attempt of in vitro differentiation screening of mutant mouse ESC lines and the successful identification of a previously uncharacterised gene, *Ahed/AU019823*, whose perturbation disrupts early haematopoiesis both in vitro and in vivo. Although precise molecular machineries by which *Ahed* is involved in haematopoiesis have remained to be elucidated, this study has unveiled a novel key regulator for haematopoietic development and sheds light on the feasibility of in vitro differentiation screening assays using a mutant ESC bank. As *Ahed* orthologues seem to be involved in the survival of immortalised cells, further analyses of the molecular mechanism underlying their function might pave the way for novel cancer therapies.

## Methods

### Ethics approval

This research complies with all relevant ethical regulations. All animal experiments were conducted in accordance with institutional guidelines and were approved by the Institutional Animal Care and Use Committee of Osaka University Graduate School of Medicine (Approval No. 30-096-013). Our manuscript abides by the ARRIVE (Animal Research: Reporting of In Vivo Experiments) guidelines for reporting of animal experiments.

### Mice

*Ahed*-floxed mice were generated in our laboratory. For conditional Cre/loxP-deletion analysis, B6.Cg-*Commd10*$^{Tg(Vav1-cre)A2Kio}$/J (*Vav1-cre*), B6.129-*Gt(ROSA)26Sor*$^{tm1(cre/ERT2)Tyj}$/J (*Rosa26-CreER$^{T2}$*), and *Tie2-cre* mice were purchased from the Jackson Laboratory. *C57BL/6-Ly5.1* (CD45.1) mice were also purchased from Jackson Laboratory. Homozygous floxed (*Ahed*$^{fl/fl}$) mice were obtained by intercrossing heterozygous (*Ahed*$^{fl/+}$) mice. *Ahed*$^{fl/fl}$ mice were mated with *Vav1-cre Ahed*$^{fl/+}$ mice to generate *Ahed* cKO mice. *Rosa26-CreER$^{T2}$ Ahed*$^{fl/fl}$ cKO mice were generated by crossing *Ahed*$^{fl/+}$ mice with *Rosa26-CreER$^{T2}$*[45]. We administered tamoxifen intraperitoneally at 1 mg/day consecutively for 5 days in postnatal 8-10-week-old *Rosa26-CreER$^{T2}$ Ahed*$^{wt/wt}$ or *Rosa26-CreER$^{T2}$ Ahed*$^{fl/fl}$ mice. Genotyping primers are shown in Supplementary Table 7. The day of vaginal plug observation was considered as embryonic day 0.5 (E0.5). Embryos were harvested on embryonic day E9.5, 10.5, 11.5, 12.5, 13.5, 14.5, 16.5, or 18.5. The FLs were dissected for flow cytometric or gross observation. Since *Vav1-Cre* females are recommended for mating in this model[46], we also mated *Vav1-Cre* female mice with *Ahed*-flox male mice and used their progeny in this study. All mice used in this study were maintained under specific pathogen-free conditions in an animal facility at Osaka University. The mice involved in this study were housed under a 12-hour light/dark cycle, with a stable temperature between 21.5 °C and 24.5 °C and a relative humidity range of 45–65%. They were provided with a standard laboratory chow diet and had ad libitum access to water. All mice used in this study were euthanized under anaesthesia.

### Maintenance and modification of ESC lines

V6.5 mouse ESCs ((C57BL/6 × 129S4/SvJae)F1)[47] and their derivatives were used for in vitro experiments. Generation of the homozygous mutant ESC clones is described elsewhere[14]. ESCs were maintained on mitomycin C-treated mouse embryonic fibroblast (MEF) feeders in serum-containing medium[48], unless otherwise indicated. cDNA for *Ahed* and its deletion mutants were subcloned into the transposon-based expression vector, pPB-CAG-IRES-Blasticidin[48,49]. These vectors were transfected into ESCs together with pCMV-hyPBase[50] using

TransFast transfection reagent (Promega). ESCs with successful integration were positively selected using $10–100\,\mu g\,mL^{-1}$ of Blasticidin. $Ahed^{m/r}$ and $Ahed^{r/r}$ revertant ESC lines were generated by the transfection of $Ahed^{m/m}$ ESCs with the pCAGGS-Flpo-IRES-puro plasmid[51] using TransFast. Three days after transfection, $1.0 \times 10^3$ ESCs were plated on a 10 cm dish. Five days later, ESC colonies were isolated and screened by PCR. PCR primers used are listed in Supplementary Table 8a–b.

## Construction of expression vectors

The pPB-CAG-IRES-Blasticidin transposon-based expression vector was used for delivery of cDNAs into mouse ESCs. First, total RNA was isolated from $Ahed^{wt/wt}$ ESCs with TRIzol reagent (Life Technologies), and then was converted to cDNA by SuperScript III First Strand Synthesis System. Second, cDNA for *Ahed* was amplified by PCR using primers AHED-F1 and AHED-R1, and digested with *Eco*RI and *Sal*I. Third, this fragment was cloned between the *Eco*RI and *Xho*I sites of the pPB-CAG-IRES-Blasticidin vector, resulting in pPB-CAG-AHED-IRES-Blasticidin.

FLAG-tagged full-length *Ahed* cDNA was PCR-amplified with primers FLAG-AHED-F and AHED-R1, using the pPB-CAG-AHED-IRES-Blasticidin vector as a template. The PCR product was digested with *Eco*RI and *Sal*I and inserted between the *Eco*RI and *Xho*I sites of pPB-CAG-IRES-Blasticidin, resulting in pPB-CAG-FLAG-AHED-IRES-Blasticidin.

To construct pPB-CAG-EGFP-AHED-IRES-Blasticidin, we connected the following three DNA fragments using an In-Fusion HD cloning kit (Takara): *EGFP* cDNA was PCR-amplified from Rosa-F3-Neo-H2B-EGFP using primers EGFP-F1 and EGFP-R1; *Ahed* cDNA amplified from pPB-CAG-AHED-IRES-Blasticidin using primers AHED-F2 and AHED-R2; and pPB-CAG-IRES-Blasticidin vector linearised with *Hpa*I and *Xho*I. In parallel, *Ahed* cDNA was amplified with primers AHED-F3 and AHED-R3, *EGFP* cDNA was amplified with primers EGFP-F2 and EGFP-R2, and the linearised pPB-CAG-IRES-Blasticidin was similarly connected using In-Fusion HD, resulting in pPB-CAG-AHED-EGFP-IRES-Blasticidin.

The *Hpa*I/*Xho*I linearised pPB-CAG-IRES-Blasticidin vector was fused with *EGFP*-*Ahed* (1–450 bp) and *Ahed* (781–879 bp) fragments amplified from pPB-CAG-EGFP-AHED-IRES-Blasticidin with primer pairs EGFP-F1 × AHED-del-R1 and AHED-del-F1 × AHED-R2, respectively, resulting in pPB-CAG-EGFP-AHED (del 451–780)-IRES-Blasticidin. Similarly, pPB-CAG-IRES-Blasticidin was fused with *EGFP* and *Ahed* (451–780 bp) fragments amplified from pPB-CAG-EGFP-AHED-IRES-Blasticidin with primer pairs EGFP-F1 × EGFP-R3 and AHED-del-F2 × AHED-del-R2, respectively, resulting in pPB-CAG-EGFP-AHED (451–780)-IRES-Blasticidin.

The *Hpa*I/*Xho*I linearised pPB-CAG-IRES-Blasticidin vector was fused with the *EGFP* fragment amplified from pPB-CAG-EGFP-AHED-IRES-Blasticidin with primers EGFP-F1 and EGFP-R4, resulting in pPB-CAG-EGFP-IRES-Blasticidin.

The *Hpa*I/*Xho*I linearised pPB-CAG-IRES-Blasticidin vector was fused with *EGFP*-*Ahed* (1–450 bp) and *Ahed* (601–879 bp) fragments amplified from pPB-CAG-EGFP-AHED-IRES-Blasticidin with primer pairs EGFP-F1 × AHED-del-R3 and AHED-del-F3 × AHED-R2, respectively, resulting in pPB-CAG-EGFP-AHED (del 451–600)-IRES-Blasticidin. Similarly, pPB-CAG-IRES-Blasticidin was fused with *EGFP*-*Ahed* (1–600 bp) and *Ahed* (781–879 bp) fragments amplified from pPB-CAG-EGFP-AHED-IRES-Blasticidin with primer pairs EGFP-F1 × AHED-del-R4 and AHED-del-F4 × AHED-R2, respectively, resulting in pPB-CAG-EGFP-AHED (del 601–780)-IRES-Blasticidin. Primers used for vector construction are listed in Supplementary Table 8b.

## OP9 system for in vitro differentiation of ESCs toward haematopoietic cells

ESCs were co-cultured with OP9 stromal cells[15,52] as described previously[53]; Minimum essential medium alpha (Life Technologies) supplemented with 10% foetal bovine serum (Equitech-Bio) and 0.05 mM 2-mercaptoethanol (Sigma-Aldrich) was used for the induction of ESCs toward haematopoietic lineages. On day 0, ESCs were

distributed onto confluent OP9 cells in a 6-well plate at a density of $1.0 \times 10^4$ cells per well. On day 4 (96–108 h after induction), $1.0 \times 10^4$ of early mesodermal cells were sorted as the Flk-1⁺ population with BD FACSAria II and then plated onto newly prepared OP9 cells in a 6-well plate. These cells were further differentiated into myeloid and erythroid cells by adding rmSCF ($50\,ng\,mL^{-1}$), rmIL-3 ($20\,ng\,mL^{-1}$), and rhEPO ($3\,U\,mL^{-1}$) into the culture medium from day 6. On day 13, non-adherent cells, i.e., haematopoietic cells, were collected, stained with monoclonal Abs against CD71 (C2) and CD11b (M1/70), and analysed by flow cytometry. The morphology of haematopoietic cells was analysed on day 13 by staining the cytospin preparations (Shandon) with May-Grünwald/Giemsa. Images were captured using BioRevo BZ-9000.

## Microscopy of in vitro cultures

$Ahed^{m/m}$ ESCs expressing either *EGFP*-fused full length *Ahed* cDNA or its deletion mutants were fixed in 4% paraformaldehyde (Nacalai Tesque) in phosphate-buffered saline (PBS), permeabilised in 0.2% Triton X-100 (Nacalai Tesque), and the nuclei were stained by $10\,\mu g\,mL^{-1}$ Hoechst 33342 (Life Technologies). For immunofluorescence, Triton X-100-permeabilised cells were incubated with 1% bovine serum albumin (Sigma-Aldrich) in PBS and stained with phycoerythrin-conjugated (PE-conjugated) anti-CD31 Ab (MEC 13.3, 1:100 dilution) for PECAM-1, and anti-FLAG Ab (M2, 1:2000) followed by Alexa Fluor 488-conjugated goat Ab to mouse IgG (1:3000) for FLAG. Nuclei were counterstained with Hoechst 33342 when necessary. Images were captured with a FV10i confocal microscope (Olympus) or a BioRevo BZ-9000 inverted microscope (Keyence).

## Construction of the targeting vector

The *Ahed* targeting vector was constructed as follows. First, as a long arm, 4.9-kb genomic lesion was PCR-amplified in 3.8-kb and 1.1-kb fragments from B6 genomic DNA with primer pairs Long-arm-F × Long-arm-LoxP-R and Long-arm-LoxP-F × Long-arm-R, respectively. These fragments were connected with LoxP-FRT-Pgk-NeoW-pA-FRT vector linearised with *Asc*I and *Pac*I using In-Fusion HD, resulting in LoxP-FRT-Pgk-NeoW-pA-FRT-Long-arm (1.1-kb-arm-LoxP-3.8-kb-arm). Next, as a short arm, 2.5-kb genomic lesion was PCR-amplified with primers Short-arm-F and Short-arm-R and was fused with LoxP-FRT-Pgk-NeoW-pA-FRT-Long-arm (1.1-kb-arm-LoxP-3.8-kb-arm) vector linearised with *Not*I and *Swa*I using In-Fusion HD, resulting in Short-arm-LoxP-FRT-Pgk-NeoW-pA-FRT-Long-arm (1.1-kb-arm-LoxP-3.8-kb-arm). This vector was linearised with *Pme*I and used for gene targeting. Primers used for this vector construction are listed in Supplementary Table 9.

## Gene targeting in ESCs

The linearised targeting vector ($25\,\mu g$) was electroporated into $1.0 \times 10^7$ KY1.1 ESCs (129S6/B6 F1 hybrid ESCs) using a Gene Pulser II (Bio-Rad) at 240 V and 500 µF. From the next day of electroporation, transfected ESCs were selected under G418 ($150\,\mu g\,mL^{-1}$) for 4 or 5 days until the appearance of individual colonies, which were separately isolated and screened for targeted homologous recombination by PCR using primer pairs PGK-L1 × Short-1R and Lox-upper × Lox-lower. Of 14 clones having homologous recombination, 3 clones were selected.

These three clones were transfected with the pMC-Cre-Pgk-puro plasmid using TransFast to generate $Ahed^{+/-}$ ESCs. The resultant clones were subjected to PCR-based screening for excision of the floxed segment using primer pairs Lox-upper × Lox-lower and Lox-upper × Lox-lower-2. Similarly, to obtain $Ahed^{fl/+}$ ESCs, three clones were transfected with pCAGGS-Flpo-IRES-puro, and the separated clones were screened for removal of the *Pgk1-neo* cassette with primer pairs FRT-upper × FRT-lower and FRT-upper × FRT-lower-2. The genomic structure of the targeted locus was further verified by PCR with primers Long-arm-check-F and Long-arm-check-R and Southern blotting. 5′- and 3′-external probes and an internal probe were PCR-amplified from B6 genomic DNA with primer pairs 5′-external-F × 5′-external-R,

3′-external-F × 3′-external-R, and Internal-F × Internal-R, respectively. PCR primers used are listed in Supplementary Table 8.

Chimeric mice with germline transmission competency were obtained from two independent *Ahed*$^{+/-}$ ESC clones by injecting them into ICR 8-cell stage embryos. Chimeras with germline potential were also generated from two independent *Ahed*$^{fl/+}$ clones by injection into (C57BL/6 × DBA/2)F2 hybrid blastocysts. In both cases, male chimeras were mated with C57BL/6 J female mice (purchased from Clea Japan) to produce heterozygous progenies.

## Chemical treatment

Mice (*Rosa26-CreER*$^{T2}$ *Ahed*$^{wt/wt}$ or *Rosa26-CreER*$^{T2}$ *Ahed*$^{fl/fl}$) were treated with intraperitoneal 1 mg per day tamoxifen (Cayman Chemical, Ann Arbor, MI) injection consecutively for 5 days to investigate *Ahed*'s postnatal role in haematopoiesis in BM. Mice were euthanised to perform assays (histological or RT-PCR) or flow cytometry plots on day 10 after tamoxifen treatment.

## Peripheral blood (PB) analyses

PB samples were obtained by cardiac puncture. PB was haemolysed using BD Pharm Lyse lysing buffer (BD Biosciences, Franklin Lakes, NJ, USA) and washed twice with PBS with 3% foetal calf serum (FCS) before antibody staining. WBCs, haemoglobin, and platelet counts in PB were examined using a blood cell analyser (KX-21; Sysmex).

## Cell isolation, analysis, and sorting

BM cells were isolated by flushing the femurs and tibias with staining buffer (PBS supplemented with 3% FCS, 100 U mL$^{-1}$ penicillin, and 100 μg mL$^{-1}$ streptomycin) by using a needle and syringe. FL and BM cells were gently filtered through a nylon screen (70 mm) to obtain a single-cell suspension. For analysing E10.5 embryos, single-cell suspensions were prepared by treating tissues with collagenase (0.125% in PBS/10% foetal calf serum (FCS)/1% penicillin/ streptomycin) for 1 h at 37 °C. Cells were blocked with an anti-CD16/32 (clone 93) antibody (BioLegend), incubated with the indicated antibodies, and resuspended in 7-AAD-containing buffer. As a negative control, we used isotype-matched antibodies in flow cytometry experiments. Flow cytometric analysis and sorting were performed using FACS-Canto or FACSAriaIIu cytometers (BD Biosciences). Doublets were excluded by FSC and SSC profiles. Lineage (CD11b, Gr1, CD3ε, CD45R/B220, and TER119)$^-$Sca-1$^+$c-Kit$^{Hi}$ (LSK) cells, LSK CD150$^+$CD48$^-$Flt3$^-$ LT-HSCs, LSK CD150$^-$CD48$^-$Flt3$^-$ MPPs, LSK Flt3$^+$IL-7Ra$^-$ LMPPs, Lin$^-$c-Kit$^{Lo}$Sca-1$^-$/$^{Lo}$Flt3$^+$IL-7Ra$^+$ CLPs, Lin$^-$Sca1$^-$c-Kit$^+$FCgR$^{Lo}$CD34$^{Hi}$ CMPs, Lin$^-$Sca1$^-$c-Kit$^+$FCgR$^{Hi}$CD34$^{Hi}$ GMPs, and Lin$^-$Sca1$^-$c-Kit$^+$FCgR$^{Lo}$CD34$^{Lo}$ MEPs were stained as previously described[54,55]. The numbers of cells in the femurs and tibias were obtained by multiplying the percentage of each population by total BM cells. Flow cytometry data were analysed using BD FACSDiva software (BD Biosciences) or FlowJo software version 9.9.6 and 10.8.1 (FlowJo, LLC).

## Colony assays

We suspended sorted LSK cells from *Ahed* cKO or control FLs or BMs in 3 mL Methocult GF M3434 (StemCell Technologies), distributed the cells into three 35 mm dishes, and incubated the cells with 5% CO$_2$ at 37 °C. After 9 days of culture, colonies were counted and classified as granulocyte colony-forming units, macrophage colony-forming units, granulocyte-macrophage colony-forming units, BFU-E, or mixed erythroid-myeloid colony-forming units according to the shape and colour under an inverted microscope.

## Stromal co-culture and limiting dilution assay

Foetal liver LSK cells were subjected to stromal co-culture for B cell differentiation, and a limiting dilution assay was performed to determine the frequency of haematopoietic progenitors, as described previously[25]. For the limiting dilution analysis, rmSCF (10 ng mL$^{-1}$),

rmFlt3-ligand (20 ng mL$^{-1}$), rmIL-7 (1 ng mL$^{-1}$), and rhEPO (3 U mL$^{-1}$) were added to the culture medium.

## In vivo transplantation Assays

For donor cell isolation, 1000 LSK cells were isolated from E14.5 foetal liver (FL) with the CD45.1$^-$CD45.2$^+$ phenotype *Vav1-cre Ahed*$^{fl/+}$ (control) or *Vav1-cre Ahed*$^{fl/fl}$ (cKO) in the first transplantation experiment (Fig. 4h), and 2000 LSK cells were isolated from the bone marrow of tamoxifen-treated *Ahed*$^{fl/fl}$ or tamoxifen-treated *R26CE Ahed*$^{fl/fl}$ mice, which exhibited the CD45.1$^-$CD45.2$^+$ phenotype in the second experiment (Fig. 5f). For recipient preparation, recipient mice were lethally irradiated and of the C57BL/6-CD45.1$^+$CD45.2$^+$ WT genotype for the first experiment, and C57BL/6-CD45.1$^+$CD45.2$^-$ WT genotype for the second experiment. Prior to transplantation, recipient mice were provided with Enrofloxacin (Elanco, Greenfield, Ind. US) at a concentration of 0.13 mg mL$^{-1}$ in drinking water for 1 week before and 4 weeks after transplantation to prevent infection. Transplantation Procedure is as follows; donor LSK cells were transplanted into the recipient mice via the orbital venous plexus injection. For both experiments, CD45.1$^+$CD45.2$^+$ or CD45.1$^+$CD45.2$^-$ WT bone marrow cells were co-transplanted as rescue cells to support engraftment. The experimental procedures are summarized in Figs. 4h and 5f, h.

## In vitro transplantation Assays

In vitro transplantation assays were conducted using bone marrow (BM) mononuclear cells from tamoxifen-treated *Ahed*$^{fl/fl}$ or tamoxifen-treated *R26CE Ahed*$^{fl/fl}$ mice, treated with 4-OHT (5 μM) for 54 hours, and subsequently transplanted into recipients (Fig. 5h). Following this treatment, 2000 LSK cells were sorted from the CD45.1$^-$CD45.2$^+$ phenotype of tamoxifen-treated *R26CE$^-$ Ahed*$^{fl/fl}$ or tamoxifen-treated *R26CE Ahed*$^{fl/fl}$ mice bone marrow and transplanted into lethally irradiated recipient C57BL/6-CD45.1$^+$CD45.2$^-$ WT mice, along with 2 × 10$^5$ CD45.1$^+$CD45.2$^-$ WT BM cells as a rescue. The 4-OHT concentration and its incubation time were finalised with reference to the results of preliminary experiments as well as previous reports.

## Whole-mount immunostaining of embryos

Whole-mount immunostaining was performed as described previously[56]. In brief, embryos were fixed for 20 to 30 min in 2% paraformaldehyde diluted in phosphate-buffered saline (PBS) on ice, and dehydrated in graded concentrations of methanol/PBS (50%, 100%; 10 min each). The yolk sac, head, limb buds and lateral body wall were removed in 100% methanol to stain around the dorsal aorta. After pre-incubation with PBS-MT (PBS containing 0.4% Triton X-100 and 1% skim milk) for at least 1 h, samples were incubated overnight with primary antibody diluted in PBS-MT. Samples were washed three times in PBS-MT throughout the next day (2 to 3 h per wash). Secondary antibodies were diluted in PBS-MT, and samples were incubated overnight. Primary antibodies were to Kit (rat, 2B8; BD Biosciences, 1:500) and CD31 (hamster, 2H8; Merck, 1:1000). Secondary antibodies were donkey anti-rat IgG-Alexa Fluor 647 (Invitrogen, 1:5000) and donkey anti-rat IgG-Cy3 (Jackson ImmunoResearch, 1:5000).

## Apoptosis analysis

ESCs were harvested by accutase (Sigma-Aldrich). Apoptotic cells were then detected using an Annexin V-FITC apoptosis kit (BioVision) according to the manufacturer's instructions. To evaluate apoptotic cells among foetal liver LSK-derived blood cells, cells were stained with APC-conjugated anti-CD11b Ab (M1/70) and BV421-conjugated anti-CD71 Ab (RI7217) in advance. CD11b$^+$ cells were analysed.

## Analysis of DNA content

DNA content of cells was evaluated as described previously[57]. The proportion of apoptotic cells (DNA content <2n) was also measured (Fig. 6c).

## RNA sequencing

For RNA-Seq on E14.5 FL or BM LSK CD48⁻ cells (Fig. 5), Total RNA was extracted from cells with an RNeasy Micro kit (Qiagen) following the manufacturer's instruction. Each cDNA was generated using a Clontech SMART-Seq HT Kit (Takara Clontech, Mountain View, CA, USA), and each library was prepared using a Nextera XT DNA Library Prep Kit (Illumina, San Diego, USA). Sequencing was performed on NovaSeq 6000 platform in a 101 + 101 base paired-end mode. Illumina RTA3 v3.4.4 software was used for base calling. Generated reads were mapped to the mouse (mm10) reference genome using TopHat v2.1.1 in combination with Bowtie2 ver. 2.2.8 and SAMtools ver. 0.1.18. Fragments per kilobase of exon per million mapped fragments (FPKMs) were calculated using Cuffdiff 2.2.1 with parameter-max-bundle-frags 50000000. (v2.2.1; http://cole-trapnell-lab.github.io/cufflinks/). Bioinformatic analyses were conducted with Ingenuity Pathway Analysis software (Ingenuity Systems; Qiagen). The heat map enables the visualisation of the differential expression data of genes categorised by their functions using the Ingenuity Knowledge Base. The colour bar indicates the z-score for each category: orange and blue squares correspond to strong predicted increase (z-score > 2) and decrease (z-score < -2), respectively. Upstream Regulator Analysis was performed to predict the activation or inhibition of transcriptional factors and growth factors based on the observed gene expression changes in the gene datasets. For RNA-Seq on triplicate differentiated populations to detect alternative splicing events (Fig. 7), RNA was converted into double-stranded cDNA libraries using the SMART-seq HT kit (Takara, Shiga, Japan) according to the manufacturer's protocol. The libraries were quantified using the Illumina Library Quantification Kit (Kapa Biosystems, Wilmington, MA, USA), and the fragment size distribution was determined using a bioanalyser. High-throughput sequencing was performed using a DNBSEQ-G400 system (MGI Tech Co., Ltd., Shenzhen, China) with 150 bp paired-end reads, which were converted into fastq files. We built the reference for STAR using the Gencode vM24 references.

After mapping reads on the genome using STAR (--twopassMode Basic −outSAMstrandField intronMotif −sjdbOverhang 100), extracted junctions using regtools junctions extract command (-a 8 -m 50 -M 500000 -s 0), and clustered them using leafcutter_cluster_regtools.py (-m50 -l 500000 −checkchrom −strand=False). Then, we quantified differentially spliced genes by leafcutter_ds.R. Visualisation was performed by leafviz.

## qRT-PCR Analysis

Total RNA was extracted using QIAzol (QIAGEN). RNA samples from LSK cells were isolated using the RNeasy Mini Kit ( > 5 × 10⁵ cells) or Micro Kit ( < 5 × 10⁵ cells; Qiagen). Reverse transcription was performed using a high-capacity RNA-to-cDNA kit (Applied Biosystems, Foster City, CA, USA). qPCR was performed using THUNDERBIRD SYBR qPCR mix (Toyobo, Osaka, Japan) on a real-time PCR 7900 HT instrument (Applied Biosystems)[14,58,59]. Data were normalised to the expression of *Gapdh* using the $2^{\Delta\Delta Ct}$ method. Information concerning the primers used for each gene is provided in Supplementary Table 8.

## Statistics and reproducibility

Data are presented as mean ± standard deviation (s.d.) unless otherwise stated. One-sided Chi-square test (Fig. 3a), Two-sided Student's *t*-test (Figs. 3e, 5c, 6a, c and 7d and Supplementary Figs. 5b, c and 6a, b), false-discovery-rate-corrected with the Benjamini and Hochberg method[60] (Supplementary Fig. 3e), one-way ANOVA with Tukey-Kramer test (Figs. 3f, 4a–c, i, j, 5e) and one-way ANOVA with Brown-Forsythe test (Figs. 1c, f, 3d, e, 4e, f, and 5d, g, i, 6b and Supplementary Figs. 2a, and 4d–f, h and 5e) were used to determine differences between datasets with Prism 7 or 8 software (GraphPad Software). The significant difference was defined as less than 0.05 of the *p* value.

*P* values are shown as *$P < 0.05$, **$P < 0.01$, ***$P < 0.001$, ****$P < 0.0001$, and NS: not significant. Representative micrographs from at least two biological replicates are shown.

## Reporting summary

Further information on research design is available in the Nature Portfolio Reporting Summary linked to this article.

## Data availability

The raw sequence data for RNA-seq analysis have been deposited at Gene Expression Omnibus (GEO) under accessions GSE218517 (RNA-seq on E14.5 FL or BM LSK CD48⁻ cells) and GSE218518 (RNA-seq on triplicate differentiated populations to detect alternative splicing events). All other data that support the findings of this study are available from the lead contact upon reasonable request. Source data are provided with this paper.

## Code availability

All original codes are in the GitHub repository [https://github.com/RITSUKONAKAI/Ahed-RNASeq-Analysis]. Any additional information required to reanalyse the data reported in this paper is available from the lead contacts upon request.

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

## Acknowledgements

We thank NPO Biotechnology Research and Development for generating chimeric mice by ESC injection. We are also grateful to Drs. Michiko Ichii, Akira Tanimura, Tomoaki Ueda, Yasuhiro Shingai, and Takayuki Ozawa for technical supports and constructive suggestions. This work was supported by the Japan Society for the Promotion of Science KAKENHI, Grants-in-Aid for Young Scientists (Grant Number 20K17379 to R.N.,

24790972 and 26860731 to M.Tokunaga) and Grants for basic research (Grant Number 21K08415 to T.Yokota). This work was further supported by JST SPRING, Grant Number JPMJSP2138 (to R.N.), Nippon Shinyaku Research Grant (to R.N.), Grant-in-Aid for Research on Development of New Drugs from AMED (to M.Tokunaga), and JST PRESTO program (to M.Tokunaga). This study was supported by the Center for Medical Research and Education, Graduate School of Medicine, Osaka University.

## Author contributions

M. Tokunaga and J.T. conceived the project. R.N., T. Yokota, and M. Tokunaga interpreted the data and wrote the manuscript. R.N. and M. Tokunaga designed, performed, and analysed experiments. T.S. and T. Yokota provided technical advice. H.S. supported mouse breeding, cell culture, and transplantation experiments. T.Yokomizo conducted immunostaining experiments with foetuses. M. Tokunaga, C.K., S.T., K.T., A.Y., J.Y., and K.H. conducted ES screening. M. Tokunaga generated *Ahed*-floxed mice. M. Takaishi performed experiments on tamoxifen-treated R26CE *Ahed* *fl/fl* mice. R.N. analysed RNA sequencing data with assistance from Y.Y., and D.O. H.W. and G.K. generated pregnant mice using frozen fertilised eggs. N.H., S.S., and J.T. supervised the study.

## Competing interests

The authors declare no competing interests.
