## [Peer Review File · Nature Communications]

REVIEWER COMMENTS

Reviewer #1 (Remarks to the Author):

In this manuscript, Nakai et al. identified a new gene named Ahead1 and demonstrated its essential role in hematopoiesis. The functional significance of Ahead1 in hematopoiesis was initially discovered using a homozygous mutant ESC bank established by this group. Using screening of ESCs with mutated 21 genes which function in hematopoietic differentiation is unknown, authors revealed that ESCs with disrupted Ahead1 showed severely compromised hematopoiesis. In vivo studies revealed that conditional deletion of Ahead1 in emerging embryonic HSCs led to anemia and impaired hematopoietic differentiation. Transcriptional profiling demonstrated that deletion of Ahead1 gene affects expression of Gata2, Lmo2, Runx1 genes and a splicing pattern for 1369 genes. Overall, this well-executed and highly significant study reveals a new gene implicated in hematopoiesis.

Comments

1. Transplantation studies with fetal liver LSK cells provide strong evidence that Ahead1 is involved in the intrinsic regulation of HSCs. However, the question remains whether the role of Ahead1 in homeostatic hematopoiesis is self-intrinsic or mediated through niche effect since Ahead1 knockout after birth causes abnormalities in multiple organs. It would be desirable to perform LSK transplantation studies with postnatal bone marrow cells from conditionally knockout Ahead1 mice, similar to those performed with FL LSK cells.

2. Ahead1 deletion doesn't affect Flk1 mesoderm formation. Does it affect the formation of hemogenic endothelium or endothelial-to-hematopoietic transition?

Minor:

1. Any accumulation of LSK cells was observed in vitro following the culture of LSK cells as shown in Fig.6?

2. Following transplantation, Ahead1 KO LSK cells fail to generate blood which explains the lack of CD45.2 chimerism. However, it would be interesting to know whether Ahead1 KO LSK cells can

engraft and persist without undergoing differentiation with the bone marrow niche of recipients. Very few CD45.2+ cells were detected in animals transplanted with cKO cells. Do they have LSK phenotype?

3. Please provide a description of cells used for cultures in Fig.6. It should be “on day 5 of culture of FL-derived LSK cells”.

Reviewer #2 (Remarks to the Author):

Nakai et al. determine that a poorly described gene they named Ahd has an essential role in embryonic and adult hematopoiesis. They identified Ahd in a screen of embryonic stem cells designed to detect genes involved in hematopoiesis. They perform a careful and well-controlled analysis of this gene's role in hematopoiesis that, although descriptive, is very well done. They also provide evidence that Ahd is involved in alternative splicing. Overall, the work is interesting and a necessary first step in defining Ahd function.

Minor comments:

1. Page 14, line 253. It is more accurate to define the Ahd mutation as a pan-hematopoietic Ahd deletion rather than Ahd deficiency, as the latter implies a germline deletion.

2. Page 14, lines 265-266. The syntax is confusing. As written, it sounds like the enriched pathways downregulated gene expression, rather than downregulated genes being enriched for genes involved in specific pathways.

3. I'm curious – why do the authors find that inflammation is higher in Ahd mutant mice (Supplemental figure 4J) whereas inflammatory pathways are downregulated (Supplemental figure 5c)? I think this is worth commenting on.

4. Page 19, line 346. I searched the paper referenced by the authors but could find nothing in the referenced study about using Vav1-Cre to delete genes.

5. Figure 3. The authors should consider including the fluor in the axis labels in panels d and f. Also, what statistical tests were used? This information should be included in all relevant figure legends.

Reviewer #3 (Remarks to the Author):

This manuscript studies mice with constitutive/germline and conditional deletion of a poorly studied gene *Ahed*. The authors identify that *Ahed* is essential for hematopoiesis and that *Ahed* is a nuclear protein. The claims made in the manuscript beyond the prior points are extremely underdeveloped. The major issues with this manuscript are as follows:

-The authors claim that *Ahed* plays a role in RNA splicing but the only analyses in that regard are RNA-seq data (performed at insufficient depth as noted below) in cells with or without *Ahed* deletion. There is virtually no global description of types, number, magnitude, or characteristics of RNA splicing events altered in association with *Ahed* deletion. The data shown are not sufficient to conclude a role for *Ahed* in RNA splicing. Any claimed effects of *Ahed* deletion on any individual splicing events could be simple indirect associations and *Ahed* could even be playing a role in mRNA expression, transcription, and/or epigenetic regulation and not RNA splicing.

-The sequencing depth performed (50 million reads per sample on average) is not sufficient for rigorous analysis of RNA splicing.

-What exact role in RNA splicing that *Ahed* plays, what RNA splicing proteins it interacts with (if any), and if it interacts with RNA would be essential for claims of *Ahed* playing a role in RNA splicing. None of these points are evaluated in the manuscript.

-The mutational frequencies of *Ahed* are very low and may not reach statistical significance based on gene size and background mutation rate.

Point-by-point responses to the reviewers' comments:

Reviewer #1 (Remarks to the Author):

In this manuscript, Nakai et al. identified a new gene named Ahead1 and demonstrated its essential role in hematopoiesis. The functional significance of Ahead1 in hematopoiesis was initially discovered using a homozygous mutant ESC bank established by this group. Using screening of ESCs with mutated 21 genes which function in hematopoietic differentiation is unknown, authors revealed that ESCs with disrupted Ahead1 showed severely compromised hematopoiesis. In vivo studies revealed that conditional deletion of Ahead1 in emerging embryonic HSCs led to anemia and impaired hematopoietic differentiation. Transcriptional profiling demonstrated that deletion of Ahead1 gene affects expression of Gata2, Lmo2, Runx1 genes and a splicing pattern for 1369 genes. Overall, this well-executed and highly significant study reveals a new gene implicated in hematopoiesis.

We sincerely thank Reviewer #1 for the thorough review and accurate understanding of our manuscript. We also appreciate the positive comments and constructive suggestions for improving the overall quality of our work. We conducted some additional experiments to address your inquiries and reported our study findings below.

Comments

1. Transplantation studies with fetal liver LSK cells provide strong evidence that Ahead1 is involved in the intrinsic regulation of HSCs. However, the question remains whether the role of Ahead1 in homeostatic hematopoiesis is self-intrinsic or mediated through niche effect since Ahead1 knockout after birth causes abnormalities in multiple organs. It would be desirable to perform LSK transplantation studies with postnatal bone marrow cells from conditionally knockout Ahead1 mice, similar to those performed with FL LSK cells.

We agree with Reviewer #1 that it is very important to clarify whether the role of *Ahed1* in homeostatic haematopoiesis is self-intrinsic or mediated through niche effect. To clarify this point, LSK transplantation studies were conducted using postnatal bone marrow cells from control (tamoxifen-treated *Ahed*^{flox/flox}) or *Ahed* conditional knockout (cKO) (tamoxifen-treated *R26CE Ahed*^{flox/flox}) mice. 2,000 control or *Ahed* cKO LSK cells of the CD45.1⁻CD45.2⁺ phenotype were transplanted along with 2×10^5 CD45.1⁺CD45.2⁻ wild-type (WT) bone marrow (BM) cells as a rescue to the lethally irradiated CD45.1⁺CD45.2⁻ WT mice (Fig. 5f). Analyses of peripheral blood leukocytes and BM cells revealed a meager

contribution of *Ahed* cKO LSK-derived haematopoietic cells three months after the transplantation (Fig. 5g and Supplementary Fig. 5e).

To exclude the possibility that *Ahed* deletion in environmental cells of the BM of tamoxifen-treated *Ahed* cKO mice might influence the haematopoietic capability of LSK cells, an additional transplantation experiment was performed in which *Ahed^{flox/flox}* or *R26CE Ahed^{flox/flox}* BM mononuclear cells were incubated with 4-hydroxytamoxifen (4-OHT) *in vitro* for 54 h; then, LSK cells were sorted and transplanted into lethally irradiated recipients (Fig. 5h). Three months after the transplantation, we observed that donor cell chimerism in the peripheral blood leukocytes was significantly lower in the 4-OHT-treated *R26CE Ahed^{flox/flox}* group (Fig. 5i). Furthermore, PCR examinations detected incompletely deleted *Ahed^{flox/flox}* band (247 bp) in the CD45.2⁺ population of peripheral leukocytes of the 4-OHT-treated group whereas a KO band at 183 bp was invisible (Supplementary Fig. 5f). These results clearly show that *Ahed* is indispensable for the haematopoietic stem/progenitor cells of the adult BM to reconstitute haematopoiesis.

These observations are now mentioned in the main text (page16, line 268 ~ page17, line 288).

2. *Ahed1* deletion doesn't affect *Flk1* mesoderm formation. Does it affect the formation of hemogenic endothelium or endothelial-to-hematopoietic transition?

We sincerely appreciate this constructive comment which can shed light on how *Ahed* is involved in haematopoietic development. We generated conditional knockout mice by crossing *Tie2-cre* with *Ahed*-floxed mice to clarify whether the deletion of *Ahed* affects haematopoietic endothelial formation and the transition from endothelium to hematopoietic cells. While these *Ahed* cKO embryos became fatal around E14.5, they were alive at E10.5, and fluorescence activated cell sorter (FACS) analysis revealed no difference in the formation of haemogenic endothelium or haemogenic clusters in the *Ahed* cKO mice compared to controls.

Furthermore, immunostaining was performed for the *Tie2-cre Ahed*-floxed E10.5 embryos. Similar to the FACS data, no significant difference in the number of haemogenic clusters was observed in the *Tie2-cre Ahed*-floxed E10.5 embryos compared to those in the control group. Next, 3D images were constructed to examine in detail whether the transition from the endothelium to haematopoietic cells was occurring. Our findings showed no obvious differences in the *Tie2-cre Ahed* cKO group compared with the control group. In summary, our study findings revealed that *Ahed* deletion did not affect the *Flk1* mesoderm formation, nor did it affect the haematopoietic endothelial formation or endothelial-to-haematopoietic transition. These results are now shown in Supplementary Fig. 4 and mentioned in the main text (page 13, line 221 ~ page 14, line 237).

Minor:

1. Any accumulation of LSK cells was observed *in vitro* following the culture of LSK cells as shown in Fig.6?

We did not detect the accumulation of LSK cells in the *in vitro* culture of *Ahed*-deficient cells. We hypothesize that *Ahed*-deficient LSK cells are unable to differentiate in this culture condition, but rather undergo apoptosis.

2. Following transplantation, *Ahead1* KO LSK cells fail to generate blood which explains the lack of CD45.2 chimerism. However, it would be interesting to know whether *Ahead1* KO LSK cells can engraft and persist without undergoing differentiation with the bone marrow niche of recipients. Very few CD45.2+ cells were detected in animals transplanted with cKO cells. Do they have LSK phenotype?

Thank you for your valuable comment. In addition to evaluating the CD45.1⁻CD45.2⁺ population in the bone marrow, we also examined the phenotype of the very few CD45.1⁻CD45.2⁺ cells. The results revealed that they did not contain a detectable number of LSK cells.

3. Please provide a description of cells used for cultures in Fig.6. It should be “on day 5 of culture of FL-derived LSK cells”.

We sincerely apologize for the insufficient explanation. According to your suggestions, we provided a correct description in the legend of Fig. 6 as follows ‘on day 5 of culture of the FL-derived LSK cells’.

Reviewer #2 (Remarks to the Author):

Nakai et al. determine that a poorly described gene they named *Ahed* has an essential role in embryonic and adult hematopoiesis. They identified *Ahed* in a screen of embryonic stem cells designed to detect genes involved in hematopoiesis. They perform a careful and well-controlled analysis of this gene’s role in hematopoiesis that, although descriptive, is very well done. They also provide evidence that *Ahed* is involved in alternative splicing. Overall, the work is interesting and a necessary first step in defining *Ahed* function.

We sincerely thank Reviewer #2 for the thorough review and accurate understanding of our manuscript. We also appreciate the positive and constructive suggestions that aim at improving the overall quality of our work.

Minor comments:

1. Page 14, line 253. It is more accurate to define the *Ahed* mutation as a pan-hematopoietic *Ahed* deletion rather than *Ahed* deficiency, as the latter implies a germline deletion.

We sincerely apologize for the insufficient explanation. According to your suggestion, we defined the *Ahed* mutation as a pan-haematopoietic *Ahed* deletion. Please see page 18, line 310.

2. Page 14, lines 265-266. The syntax is confusing. As written, it sounds like the enriched pathways downregulated gene expression, rather than downregulated genes being enriched for genes involved in specific pathways.

Thank you for your careful reading. We agree that the syntax was confusing; thus, we modified the indicated statement as follows:

'Comparative transcriptome analysis was performed to extract the differentially expressed genes (DEGs); the study findings revealed that the genes involved in the immune response pathway were enriched and downregulated in the *Ahed* cKO FL LSK cells, compared with those in the controls.' Please see page 18, lines 321-324.

3. I'm curious – why do the authors find that inflammation is higher in *Amed* mutant mice (Supplemental figure 4J) whereas inflammatory pathways are downregulated (Supplemental figure 5c)? I think this is worth commenting on.

Thank you for the comments. Supplementary Fig. 5 shows that the inflammatory pathway that is downregulated in *Ahed* deficiency is based on the analyses of haematopoietic cell-specific knockout FL or BM LSK cells. Data from mice with systemic *Ahed* knockout are shown in Supplementary Fig. 4j. The skin and jejunum tissues of the systemic *Ahed* knockout mice became atrophic presumably due to enhanced apoptosis, which might not be associated with inflammation.

4. Page 19, line 346. I searched the paper referenced by the authors but could find nothing in the referenced study about using Vav1-Cre to delete genes.

Thank you for your careful review of all the manuscript details. We sincerely apologize for the confusion. The article with citation number '34' describes the latter part of the sentence 'authentic HSC-derived CMPs, MEPs, and GMPs account for less than 20% of the total, even at E16.5'. We referenced another relevant study to describe the notion that EMPs and authentic HSCs are generated from distinct origins. Please see page 20, lines 354-356.

5. Figure 3. The authors should consider including the fluor in the axis labels in panels d and f. Also, what statistical tests were used? This information should be included in all relevant figure legends.

Thank you for the suggestion. We added the information regarding fluor and the used statistical methods to all the relevant figure legends.

Reviewer #3 (Remarks to the Author):

This manuscript studies mice with constitutive/germline and conditional deletion of a poorly studied gene *Ahed*. The authors identify that *Ahed* is essential for hematopoiesis and that *Ahed* is a nuclear protein. The claims made in the manuscript beyond the prior points are extremely underdeveloped. The major issues with this manuscript are as follows:

-The authors claim that *Ahed* plays a role in RNA splicing but the only analyses in that regard are RNA-seq data (performed at insufficient depth as noted below) in cells with or without *Ahed* deletion. There is virtually no global description of types, number, magnitude, or characteristics of RNA splicing events altered in association with *Ahed* deletion. The data shown are not sufficient to conclude a role for *Ahed* in RNA splicing. Any claimed effects of *Ahed* deletion on any individual splicing events could be simple indirect associations and *Ahed* could even be playing a role in mRNA expression, transcription, and/or epigenetic regulation and not RNA splicing.

We are truly thankful for your careful reading and critical comments regarding our manuscript. The editor agrees that the data shown are not sufficient to conclude a role for *Ahed* in RNA splicing. We also agree with your comments, 'Any claimed effects of *Ahed* deletion on any individual splicing events could be simple indirect associations, and *Ahed* could even be playing a role in mRNA expression, transcription, and/or epigenetic

regulation, and not RNA splicing'. We removed the claims regarding RNA splicing from the title and abstract; we also toned the conclusion down in the whole manuscript text. Kindly find below the point-by-point responses to your comments.

-The sequencing depth performed (50 million reads per sample on average) is not sufficient for rigorous analysis of RNA splicing.

Regarding the depth of reads in the splicing analysis, the Illumina website states that experiments that need information regarding alternative splicing typically require 30–60 million reads per sample; moreover, this read range encompasses the most published RNA-Seq experiments for mRNA/whole transcriptome sequencing (https://knowledge.illumina.com/library-preparation/rna-library-prep/library-preparation-rna-library-prep-reference_material-list/000001243). We checked the accuracy and reliability of this read depth by making saturation plots for the number of splicing junctions detected in each sample. As shown in the below figure, the detected number of known splicing junctions became saturated when the read depth was over 30 million. Therefore, we followed Illumina's guideline to conduct this analysis and set the read depth at 50 million reads. For your convenience, we prepared a dropbox link in which all the saturation plots are stored as follows:

(<https://www.dropbox.com/sh/dohewzhs7ilg1fi/AAA9mZvgtSrRN0x1RLUqHcPra?dl=0>)

We agree with the Reviewer #3 that detailed analysis of RNA splicing including novel junctions would need more read depth; however we respectfully claim that our present data with the depth of 50 million reads are robust enough to conclude the abnormal frequency and pattern of known splicing junctions.

-What exact role in RNA splicing that *Ahed* plays, what RNA splicing proteins it interacts with (if any), and if it interacts with RNA would be essential for claims of *Ahed* playing a role in RNA splicing. None of these points are evaluated in the manuscript.

Kindly find below another study that was conducted and submitted by our co-authors.

Although it is still undergoing peer review, it is available as a preprint at:

<https://assets.researchsquare.com/files/rs-3234334/v1/c544e979-dfba-4119-b164-61a01f4384f3.pdf?c=1699278362>. The study results show that *Ahed* is likely to be associated with some spliceosome proteins such as PRPF8, SF3B1 and RBM39 in HeLa cells. While it would be challenging for us to prove the interaction of *Ahed* with other proteins or RNA in the extremely small cell population like haematopoietic stem cells, we aim at addressing these issues in our future studies.

-The mutational frequencies of *Ahed* are very low and may not reach statistical significance based on gene size and background mutation rate.

We consider that mutational frequencies of *Ahed* in human acute myeloid leukaemia have not been accurately evaluated. As no pathogenic genome abnormality can be determined in a substantial number of patients with acute myeloid leukaemia at this stage, we believe that the recurring detection of *Ahed* missense mutation in acute myeloid leukaemia is very important. Reaching statistical significance or not is necessarily insignificant as data on pathogenic genome abnormality in those patients remain insufficient.

REVIEWERS' COMMENTS

Reviewer #1 (Remarks to the Author):

All my criticisms are adequately addressed by the authors. I noticed a type in paragraph title: line 221. It should be unessential or not required. Otherwise, the title contradicts the conclusion.

Overall this is very solid work providing a comprehensive analysis of the role of new gene Ahead1 in hematopoiesis.

Reviewer #1 (Remarks on code availability):

I am not sure what is supporting code. If it related to software, I was not able to find it.

Reviewer #2 (Remarks to the Author):

The authors have adequately addressed the reviewers' comments.

Reviewer #3 (Remarks to the Author):

The manuscript has been revised appropriately. The title of the manuscript has been changed and is now very awkwardly written. The authors remove the phrase "throughout the life" from the title to improve the grammar of the title.

Point-by-point responses to the reviewers' comments:

Reviewer #1 (Remarks to the Author):

All my criticisms are adequately addressed by the authors.

We are relieved that you are of the opinion that we have adequately addressed all of your criticisms.

I noticed a typo in paragraph title: line 221. It should be unessential or not required. Otherwise, the title contradicts the conclusion.

The paragraph title was clearly an error. We sincerely apologise for the typo and thank you for bringing this to our attention. In the final draft, we have changed the description to 'unessential' as you suggested.

Overall this is very solid work providing a comprehensive analysis of the role of new gene Ahead1 in hematopoiesis.

We are grateful for your kind words. With this analysis, we have been able to show the molecule Ahead1 to the world for the first time. We will continue to research tirelessly to expand the possibilities of this molecule. We would like to thank you for your review, which has improved this paper. Thank you very much.

Reviewer #1 (Remarks on code availability):

I am not sure what is supporting code. If it related to software, I was not able to find it.

We apologise that our guidance in the revised version was inadequate and made it difficult for you to understand. Following this report of acceptance in principle, we have deposited the code used for this analysis here (<https://github.com/RITSUKONAKAI/Ahed-RNASeq-Analysis>) and made it publicly available. We have provided 'Data Availability' and 'Code Availability' as separate sections and updated the 'Code Availability' information, please check.

Reviewer #2 (Remarks to the Author):

The authors have adequately addressed the reviewers' comments.

We are pleased that you feel we have adequately addressed all of the reviewers' comments.

Thank you for your review. It has helped us improve our paper. We really appreciate it.

Reviewer #3 (Remarks to the Author):

The manuscript has been revised appropriately. The title of the manuscript has been changed and is now very awkwardly written. The authors remove the phrase "throughout the life" from the title to improve the grammar of the title.

We have removed the phrase 'throughout the life' from the title as you suggested. Your discussion and comments, especially on RNA splicing, have renewed this paper. Thank you for your reviewing it for us.